# Study of the Hydrothermal-Catalytic Influence on the Oil-Bearing Rocks of the Usinskoye Oil Field

**Irek I. Mukhamatdinov** [1,*] , **Artem V. Lapin** [1] , **Rezeda E. Mukhamatdinova** [1] , **Aydar A. Akhmadiyarov** [1] ,
**Boudkhil Affane** [1] , **Dmitriy A. Emel'yanov** [1] , **Olga V. Slavkina** [2] **and Alexey V. Vakhin** [1]

1 Institute of Geology and Petroleum Technologies, Kazan (Volga Region) Federal University,
18 Kremlyovskaya Str., 420008 Kazan, Russia
2 LLC «RITEK», 85 Lesogorskaya Str., 400048 Volgograd, Russia
* Correspondence: iimuhamatdinov@gmail.com

**Abstract:** In this work, a synthesis of an oil-soluble iron-based catalyst precursor was carried out and its efficiency was tested in a laboratory simulation of the aquathermolysis process at different temperatures. The rocks of the Usinskoe field from the Permian deposits of the Komi Republic, obtained by steam-gravity drainage, and the iron-based catalyst precursor, as well as the products of non-catalytic and catalytic aquathermolysis, were selected as the object of study. As a result, it was found that the content of alkanes in the samples after thermal steam treatment (TST) at 300 °C increased 8-fold compared to the original oil, and the content of cycloalkanes in the sample with the catalyst increased 2-fold compared to the control experiment. This may indicate that not only the carbon-heteroatom bonds (C-S, N, O) but also the C-C bonds were broken. It also shows that increasing the iron tallate concentration at TST 300 °C leads to a decrease in the molecular mass of the oil compared to the control experiment. According to SEM, the catalyst is nanodisperse particles with a size of ≈60–80 nm, which are adsorbed on the rock surface, catalyst removal occurs at a small scale.

**Keywords:** aquathermolysis; catalyst precursor; thermal steam treatment; iron tallate; viscosity; SARA; GC-MS; molecular mass; SEM; TG-DSC

## 1. Introduction

In the literature, the change in oil properties under the influence of thermal factors in the environment of water steam and rock minerals is called aquathermolysis and is considered a model for the transformation processes of hydrocarbons in the formation using thermal methods to enhance oil recovery [1–3].

Compounds based on transition metals are the most popular and studied compounds for catalytic aquathermolysis [4]. Although significant progress was made in the development of transition metal-based catalysts for aquathermolysis, the main challenges lie in reducing the cost of their production and application.

The works [5,6] studied hydrothermal effects on the properties of deposits where oil-saturated sandstones were enriched with iron-containing clays (cold lake field) heated in the presence of steam at 150–250 °C for several weeks. The results showed that hydrothermal reactions in the clay minerals strongly degrade the properties of the deposit and lead to a decrease in porosity and permeability.

The authors in [7] performed physical modeling of aquathermolysis of highly viscous oil in the presence of mineral constituents of carbonate rocks consisting of dolomite and calcite. On the basis of group composition analysis and infrared spectroscopy, it was found that in the presence of calcite the content of resins and asphaltenes is significantly reduced due to the destruction of the least stable C-S-C bonds. On the basis of gas chromatography, it was found that the thermal transformation of bitumoids led to an increase in the proportion of hydrocarbons in the homologous series C19–C30. Moreover, the analysis of geochemical

coefficients showed a high maturity degree of the organic matter of thermally converted bitumen in the model system with dolomite. However, in the presence of rock, the catalytic potential in terms of heavy oil accumulation could not be fully demonstrated.

At the same time, scientists discovered in [8] that although minerals have a catalytic activity under thermal influence in the environment of water steam, due to the hydrogen donor, along with an increase in viscosity and density, the chemical composition of the residual oil changes. Namely, the share of high-molecular-mass components of asphaltenes increases significantly.

Numerous scientists investigate the reactions of aquathermolysis of organic matter and catalytic systems that include various hydrogen donors [9–11]. This is related to the possibility of intensifying cracking and hydrogenolysis reactions and, as a consequence, increasing the degree of viscosity reduction due to breaking C-S bonds in high-molecular-mass components. In addition, the catalysts facilitate the release of hydrogen from naphthenic and aromatic compounds that are part of the oil [12,13].

According to [14,15], the hydrogen content can significantly affect the composition of the reaction products formed during hydrocracking in a heated tank. Moreover, pyrolysis of heteroatomic compounds and polycyclic aromatic hydrocarbons in the absence of hydrogen leads to the formation of hydrocarbon mixtures containing double bonds and radicals.

The use of hydrogen enables the reactions of hydrogenolysis, hydrogenation and hydrocracking, which reduce the formation of double bonds and the polymerization of the resulting hydrocarbons [16].

In [17] the conversion of heavy oil in sandstone formations in the presence of water steam, hydrogen donor, and oil-soluble precursor catalyst based on iron transition metal was studied. According to the results of determining the chemical composition of the group, it was found that the use of the catalyst (2.0 wt.% bitumoid in the rock) ensures the conversion of resins by 24% at 250 °C. In addition, the use of the catalyst allows the transformation of asphaltenes at 300 °C, resulting in a reduction of asphaltenes to almost «traces» at this temperature. Enhancement of the destructive process of dissolution of heteroatomic bonds in the molecules of resins and asphaltenes contribute to the generation of light hydrocarbon fluids.

In work [18], Chinese scientists developed an oleic acid-modified $NiFe_2O_4$ nanocatalyst for heavy oil refining. The destruction of long-chain structures and hydrogenation of unsaturated aromatic compounds were observed under the given conditions of catalytic aquathermolysis. Oleic acid molecules are coordinately bound to the surface of $NiFe_2O_4$ nanoparticles, which can prevent nanoparticle aggregation and lead to minor structural and morphological changes of the catalyst.

Experimental results [19] prove that the combination of ultrasound and a catalyst has a synergistic effect, which causes a decrease in viscosity and improvement of the quality of heavy oil.

Catalysts representing Mo, W, C and Ni alloys, Cu and Fe nano-oxides, Ni-helates, Fe-based nanoparticles and $Cu^{2+}$ and $Fe^{3+}$ toluene sulfonic acid complexes are due to good efficiency in aquathermolysis reactions [20].

Transition metal carboxylates (Ni, Co, Fe) [21] were used to upgrade heavy oil under reservoir conditions. In addition, studies of catalytic aquathermolysis were carried out at temperatures of 250 °C, 300 °C and 350 °C in the presence of a naphthenoaromatic hydrogen donor. Moreover, it was found that at temperatures of 300 °C and 350 °C significant changes in oil content occur, accompanied by an increase in light fractions and a decrease in high molecular mass hydrocarbons.

The use of iron (III) tris-acetylacetonate complex had a significant effect on the aquathermolysis of heavy oil and resulted in a more extensive conversion of high molecular mass components than the non-catalytic process [22].

In experiments [23] on catalytic aquathermolysis in the presence of the rock-forming mineral kaolin using an oil-soluble iron carboxylate and a proton donor, tetralin, a decrease

in the viscosity of heavy oil was observed. The studies were carried out at temperatures of 250, 300 and 350 °C.

Autoclave studies of heavy oil showed high efficiency of $NiFe_2O_4$-based nanocatalyst [24]. As a result, in addition to a decrease in viscosity, a decrease in sulfur content by more than 20% was observed. Analysis of the spent catalyst by X-ray photoelectron spectroscopy also showed that an iron sulfide (FeS) phase was formed.

The study [25], addresses the change in phase composition under the influence of the catalyst and TST, where the decrease in sulfur and maghemite content indicates the efficiency of catalytic aquathermolysis.

In the work [11], a series of experiments were carried out at a pressure of 3 MPa for 36 h, at temperatures from 300 to 340 °C, in an elementary model with standard reservoir parameters, where hydrogen and a catalytic suspension were injected into a heavy oil conversion tank in a porous medium. The studies proved that the catalyst nanoparticles affect the viscosity reduction, thereby increasing its performance.

Thermal degradation of high molecular mass oil components in the temperature range 325–425 °C and in the presence of quartz promotes the formation of lower molecular mass substances. High molecular mass products form mainly with calcite. However, in the presence of montmorillonite and quartz, the proportion of low molecular mass compounds increases, especially aromatic compounds and limiting hydrocarbons [26].

In the work [27], a catalyst based on $Fe_3O_4$ particles was synthesized and its efficiency was tested at different concentrations after laboratory simulation of the aquathermolysis process. From the obtained data we can conclude that at temperatures of 200 and 300 °C and a concentration of 1.0% magnetite suspension stands out from all the results, showing the best efficiency of enrichment of oil Ashalchinskoye field. Taking into account the formation temperature at which the Ashalchinskoye field developed by steam-heat treatment (180–200 °C), the temperature of 200 °C is quite feasible. Taking into account economic factors, the increase in the concentration of magnetite when using this catalyst in areas with high-viscosity oils and natural bitumen, compared to oil-soluble catalyst precursors, will have little impact on pricing (production + implementation of technology).

Further, after aquathermolysis at 300 °C:

- the increase in the amount of gases released was greater than at 200 and 250 °C;
- the degree of viscosity reduction increased by 53% with a catalyst with a concentration of 1.0 wt.% compared to the original oil;
- After aquathermolysis at 200 and 250 °C:
- About the same amount of gas released at 200 and 250 °C;
- the degree of viscosity reduction at a catalyst concentration of 1.0 wt.% increased by 11% at 200 °C and by 15% at 250 °C in the presence of the catalyst compared to the original sample;
- Increase in saturated hydrocarbon content by 28% and decrease in resin content by about 30% at magnetite suspension concentration of 0.2 wt.% at TST 250 °C compared to the control experiment.

The aim of the work is to study the effect of iron tallate on the in situ transformation of extra-viscous oil of the Usinskoye field.

Scientific novelty consists of the fact that the studies justify the possibility of wide application of catalytic compositions in porous mineral media of carbonate reservoir rocks.

## 2. Results and Discussion

Study of the structural composition of rocks by X-ray powder diffractometry.

Figure 1 shows the results of the phase composition of the original rock sample after extraction according to X-ray diffraction analysis.

According to the results obtained by X-ray structural analysis of extracted oil-saturated rock from the Usinskoye field, it was found that the sample consists of ~100% calcite ($CaCO_3$), i.e., the rock of the Usinskoye field is carbonate.

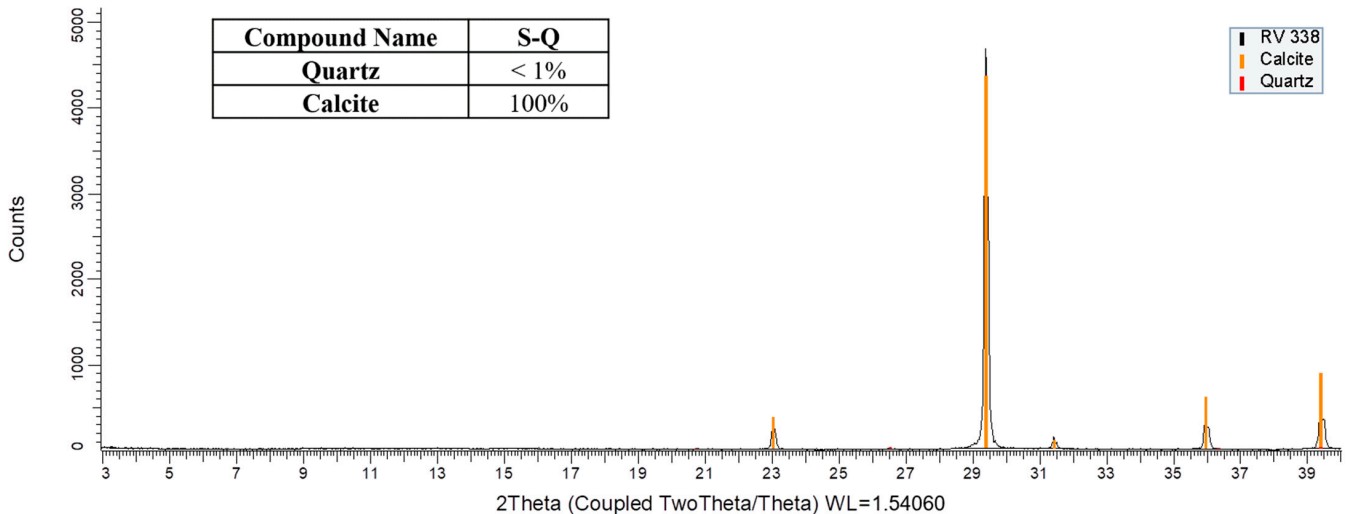

**Figure 1.** X-ray diffraction analysis of the original rock.

### 2.1. Changes in the Composition of Gases

The figures show the content and composition of gases (Figures 2–6) per 1 ton of oil, depending on the temperature of TST and the presence of iron tallate.

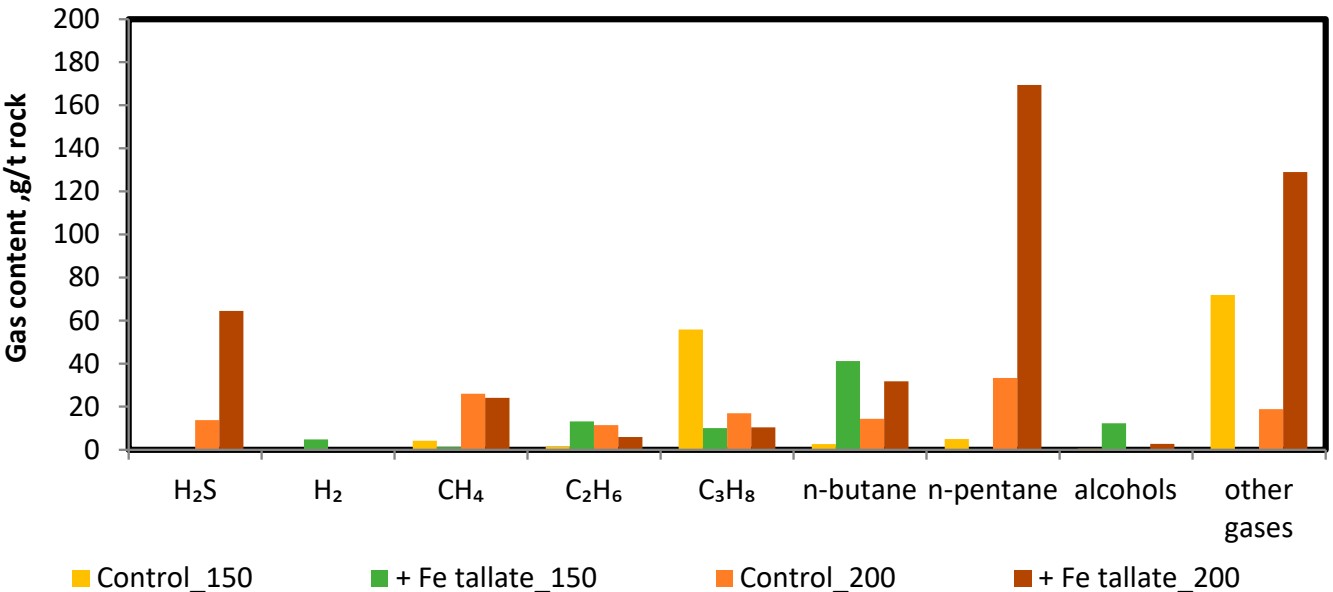

**Figure 2.** Content and composition of gases (individual gases, alcohols and other gases) depending on TST temperature and the presence of iron tallate.

In the presented data, the TST temperature of 150 °C has an insignificant effect on the increase in the gas phase content. Catalytic aquathermolysis at 150 °C in the presence of iron tallate revealed a decrease in olefins, dienes, cycloalkanes and aromatic compounds compared with the control experiment, which probably indicates the hydrogenation of obtained compounds during cracking as a result of hydrogen transfer with nefras as a hydrogen donor.

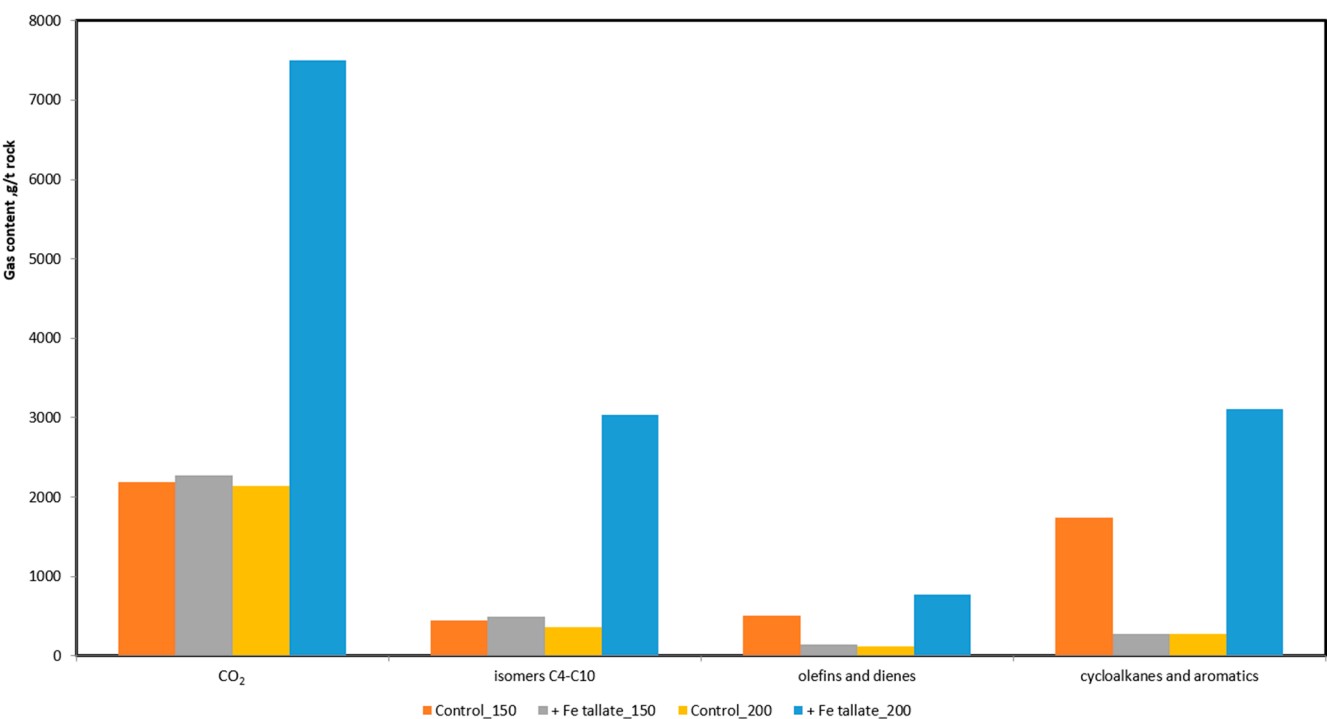

**Figure 3.** The content and composition of gases ($CO_2$, C4-C10 isomers, olefins and dienes, cycloalkanes and aromatic HCs) depending on TST temperature and the presence of iron tallate.

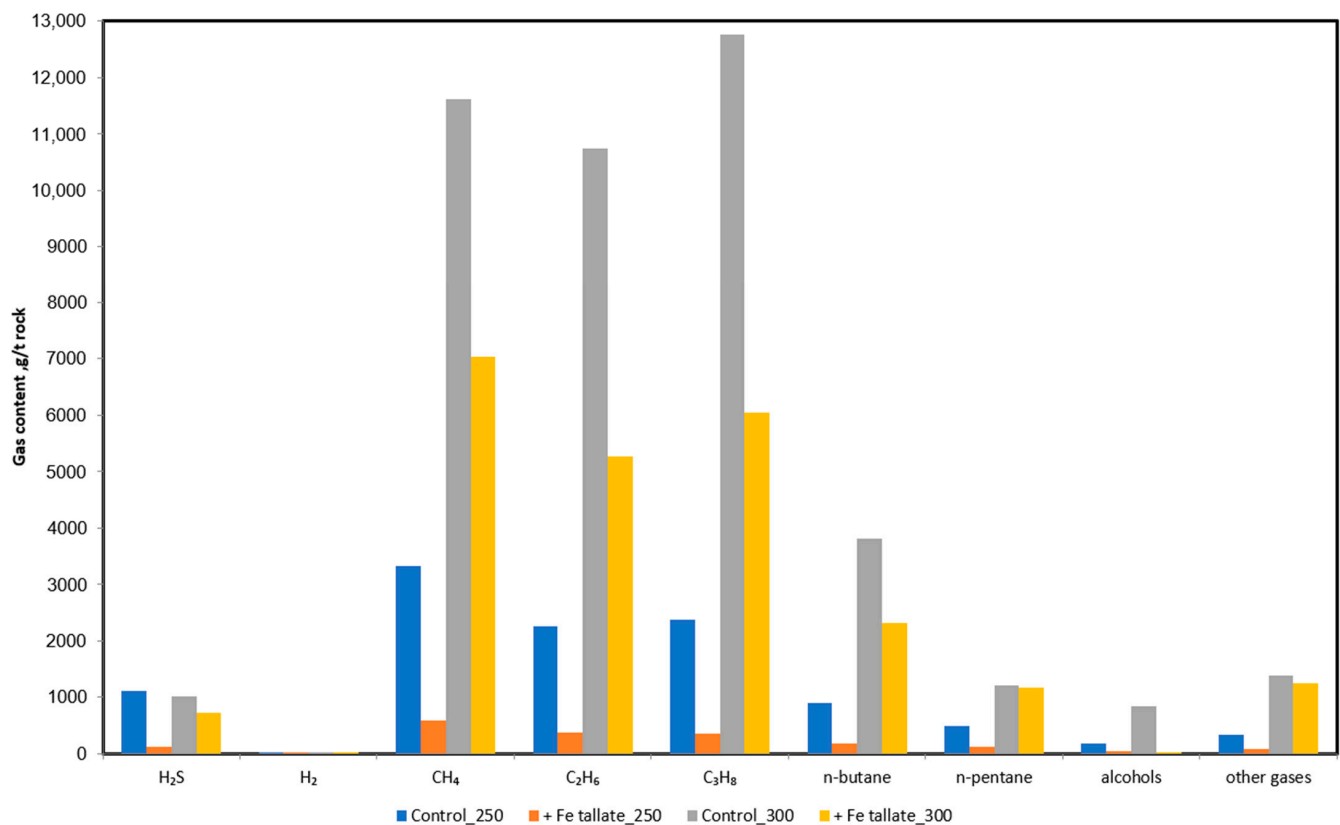

**Figure 4.** Content and composition of gases (individual gases, cycloalkanes and aromatic HCs, alcohols and other gases) depending on TST temperature and the presence of iron tallate.

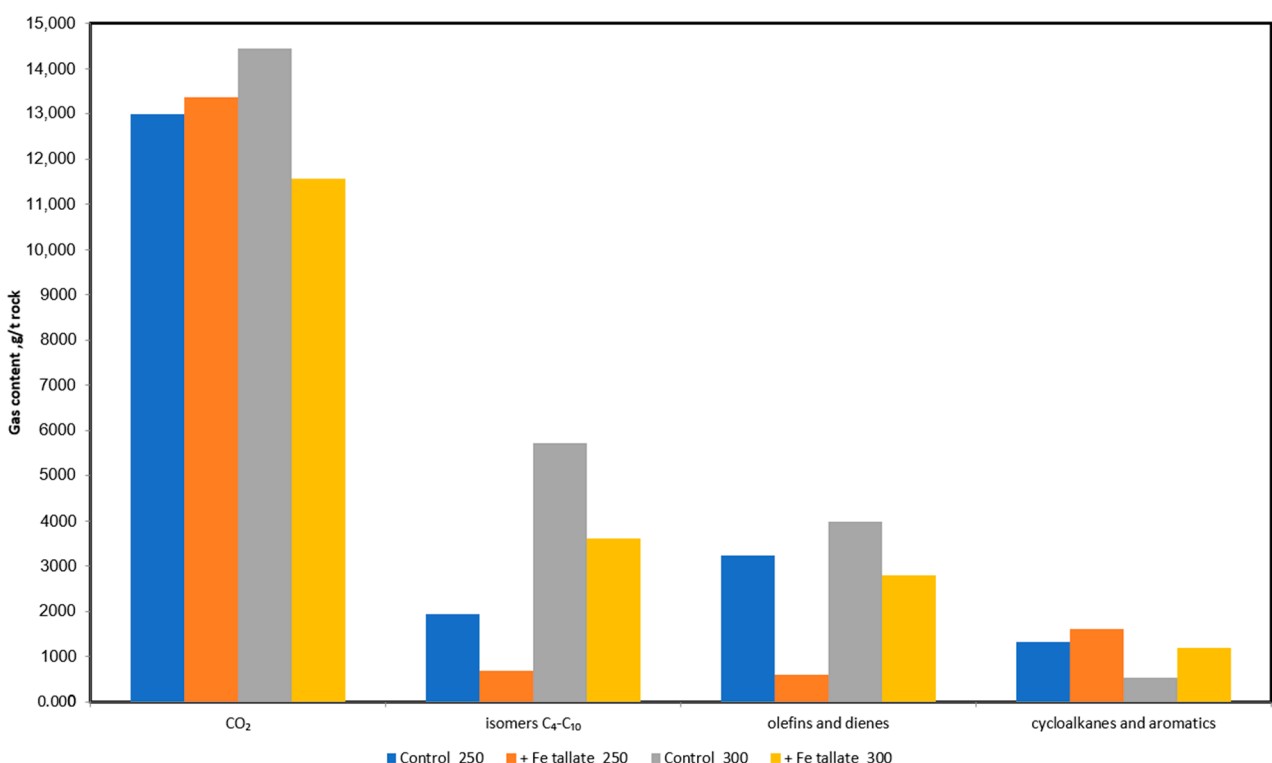

**Figure 5.** The content and composition of gases ($CO_2$, C4–C10 isomers, olefins and dienes) depending on TST temperature and the presence of iron tallate.

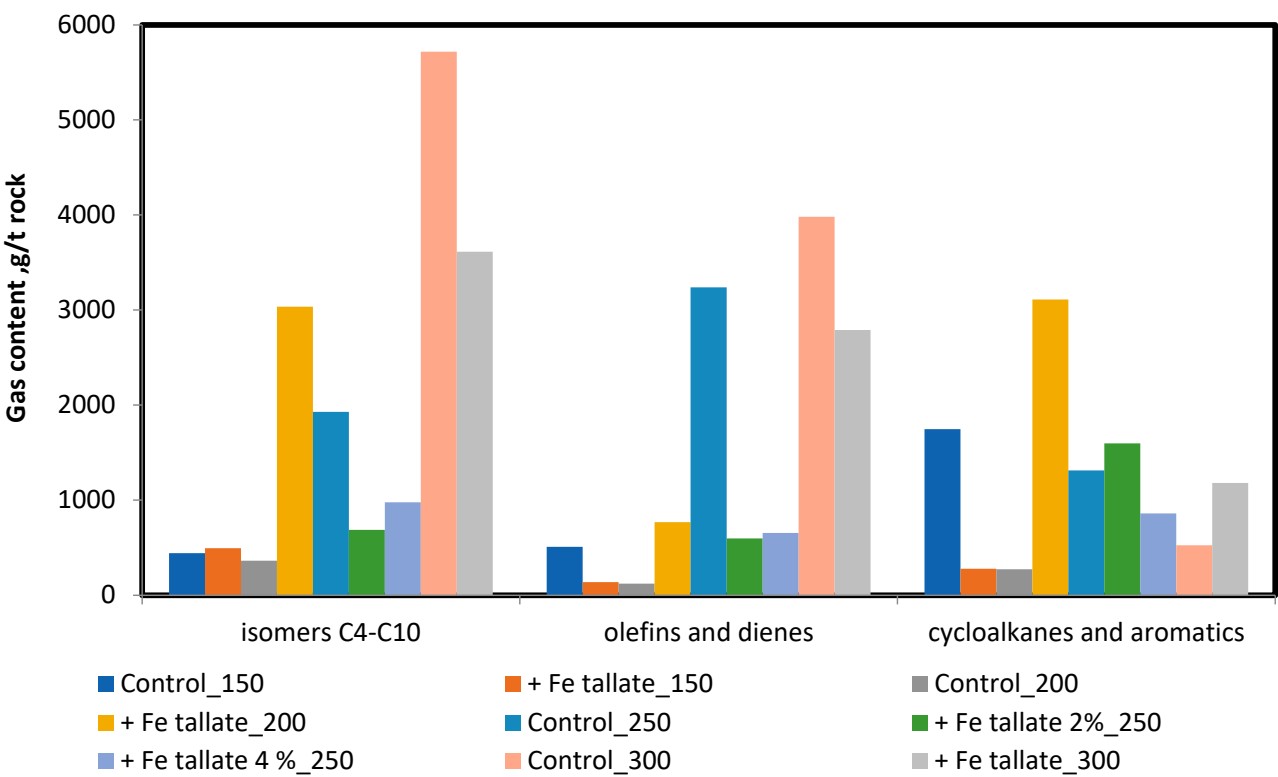

**Figure 6.** Gas content and composition (mixture of C4–C10 isomers, olefins+dienes and cycloalkanes+aromatic HCs) depending on TST temperature and presence of iron tallate.

During catalytic aquathermolysis at 200 °C in the presence of iron tallate, the $CO_2$ content increased significantly as a result of decarboxylation reactions forming lower

molecular mass hydrocarbons. The resulting carbon dioxide was involved in reducing the viscosity of the oil and contributed to the chemical composition of the group [28–30]. In addition, the hydrogen donor reduces the formation of aromatic hydrocarbons in the gas phase and the polymerization of the resulting hydrocarbons. TST at 200 °C and in the presence of a catalyst leads to an increase in the content of olefins and dienes as a result of cracking of the side chains of resinous-asphaltene substances (RAS).

At a temperature of 250 °C, the presence of iron thallate led to a decrease in the hydrogen sulfide content, probably it is involved in the formation of the sulfide form after the decomposition of the catalyst precursor. The increase in concentration affects the formation of C4–C10 isomers. This reflects the increase in the degree of oil conversion and confirms the radical mechanism of the processes taking place [12].

Catalytic aquathermolysis at 300 °C with ferrous thallate led to an increase in the content of cycloalkanes and aromatic hydrocarbons in the gas phase, which is the result of breaking longer hydrocarbon chains to form hydrocarbons with a smaller number of carbon atoms.

### 2.2. Changes in Rheological Properties

Oil of Usinskoye field is a non-Newtonian viscoelastic liquid. Viscosity of the original oil of the Usinskoye field at 25 °C is 188,559 mPa·s. Figure 7 shows the viscosity-temperature properties of non-catalytic and catalytic aquathermolysis oils at 150, 200, 250 °C, as well as the original bitumen.

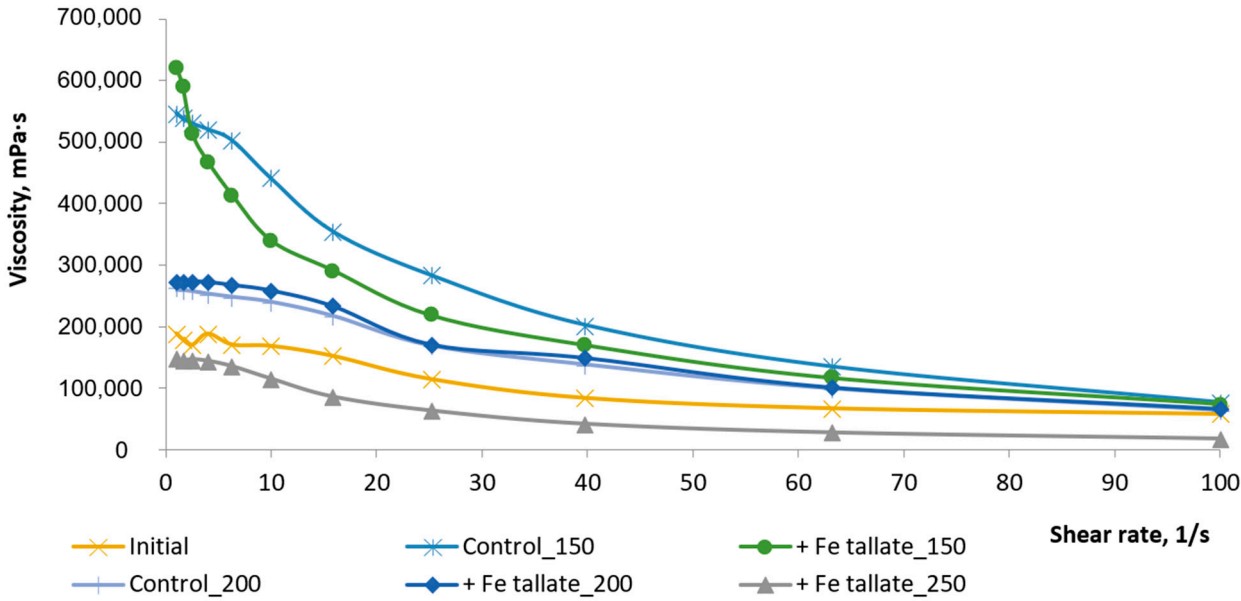

**Figure 7.** Viscosity-temperature characteristics of control experiment and oils after TST with iron thallate.

Viscosity of noncatalytic and catalytic aquathermolysis oil at 150 and 200 °C clearly increases as a result of increasing the content of resins and coking. It can be concluded that the enrichment of high-viscosity oil of the Usinskoye field at these temperatures is not effective.

The oil viscosity after TST at 250 °C with the catalyst decreased by about 1.4–3.3 times at the corresponding shear rates compared to the original sample due to an increase in the depth of conversion of high-molecular mass components of heavy oil, as well as weakening of intermolecular interactions of aggregative combinations due to the appearance in the system of a hydrogen donor, which prevents recombination of formed radicals, which provides irreversible changes in oil composition and stable value of oil viscosity over time [31,32].

Figure 8 shows the dependence of the viscosity of oils of the control experiment and the sample after TST in the presence of the catalyst after hydrothermal-catalytic effect at 300 °C. Viscosity of oil after TST at 300 °C with the catalyst decreased by about 2.5 times compared to the control experiment at the same temperature with a concentration of 2%.

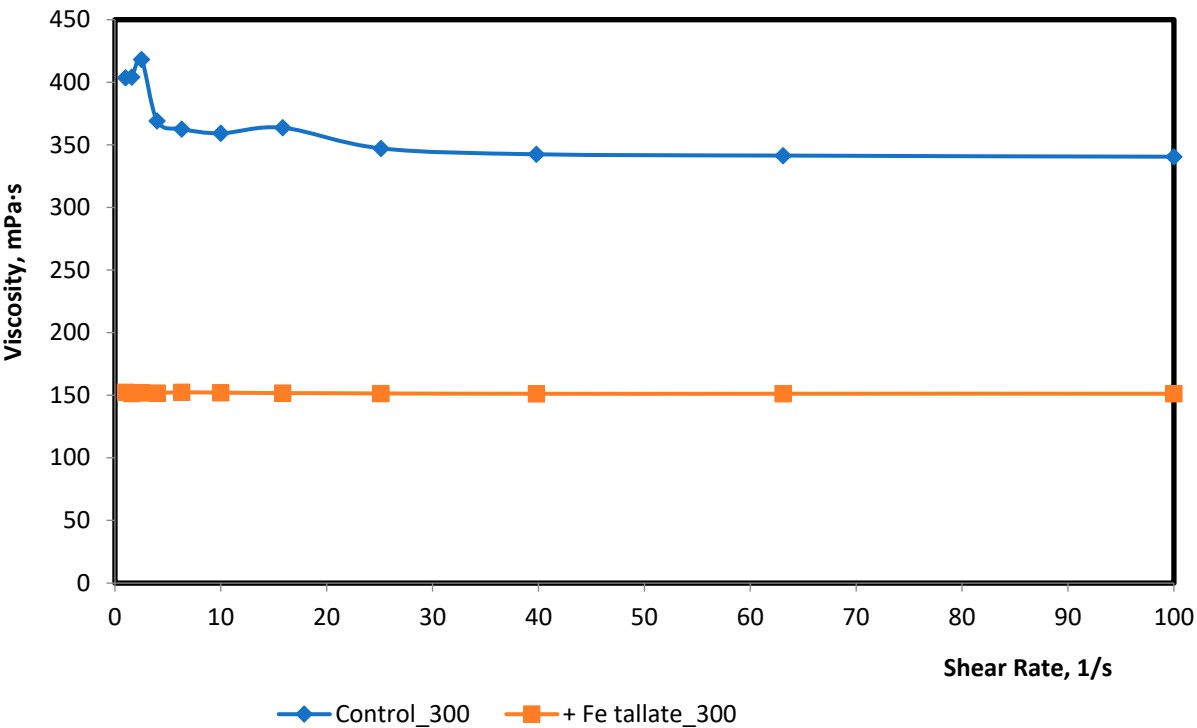

**Figure 8.** Viscosity-temperature characteristics of control experiment and oils after TST with iron tallate.

Viscosity decreases significantly as a result of chemical conversion of resins and asphaltenes, which is confirmed by the results of component analysis. It should be noted that the experiments in the autoclave were carried out for 24 h. Perhaps further increasing the duration of exposure will achieve a greater reduction in viscosity and complexity of the gas phase composition [33].

*2.3. Changes in Component Composition*

Currently, the study of the chemical composition of oils, representing a complex structural unit with a large number of chemical compounds of different composition and structure, is carried out almost all over the world, mainly using reliable methods of their separation into groups, in particular by SARA-analysis [34].

The high content of asphalt-resinous substances is characteristic of near-surface sediments in the conditions of near-surface weathering. They are enriched with hydroxyl and ester bonds, which are unstable and subject to strong degradation when exposed to heat [35,36]. Figure 9 shows the component composition by SARA method of noncatalytic and catalytic aquathermolysis oils at different temperatures.

Thus, the original oil of the Usinskoye field contains a large number of heavy components and according to the total content of resins and asphaltenes belongs to high-saline, according to the classification of Uspensky V.A. [37]. The results of the group chemical composition show that in comparison with the original oil and the control experience during the catalytic aquathermolysis process during 24 h at 300 °C with the catalyst (2%) the saturated hydrocarbons content (HC) increases by 10% (comparison with the control experience) and the aromatic hydrocarbons content increases by 5% (comparison with the original oil). Probably the catalytic agent is involved in thermo destructive decomposition

of aliphatic side chains of asphaltene molecules. Compact secondary asphaltenes with lower molecular mass are formed.

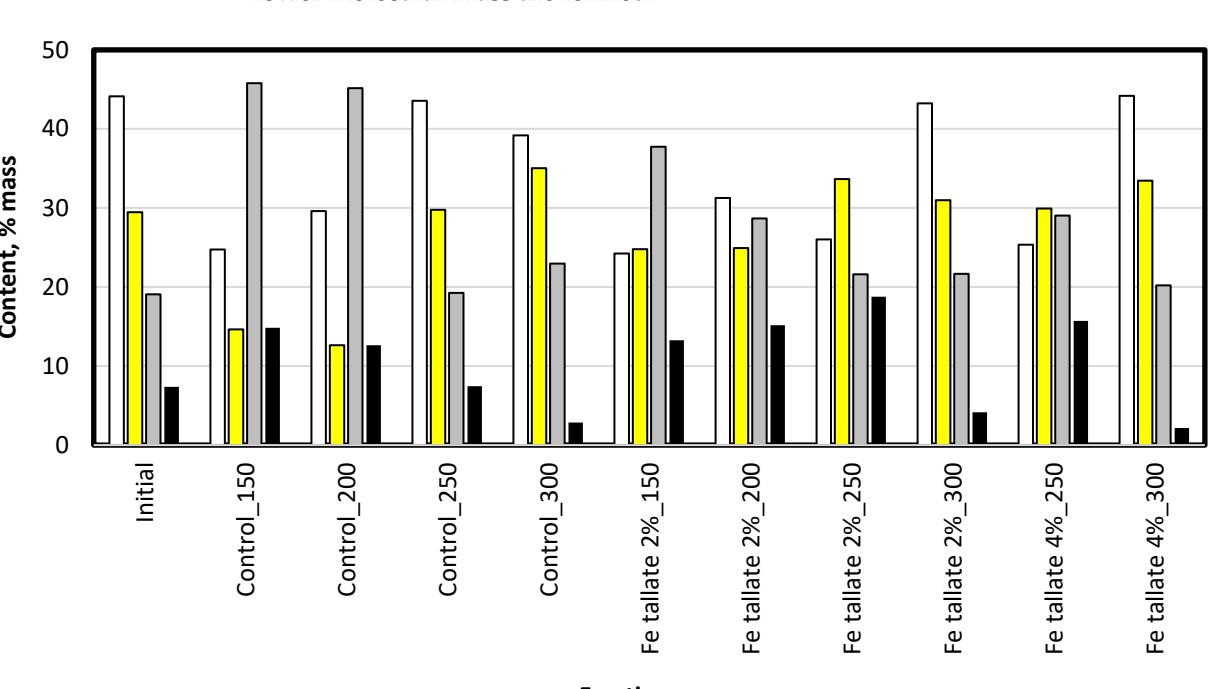

**Figure 9.** Component composition of the oil of the control experiment and oils after TST with iron tallate.

By increasing the concentration of iron tallate by 2 times in the reaction mixture at 300 °C—TST resulted in a 48% decrease of asphaltene content compared to the 300 °C $Fe_{2\%}$ experience, by 71% compared to the original oil.

According to the results of the component composition of the oil after TST 150 and 200 °C, the coking and resinization occurs, which leads to an increase in the content of high-molecular-mass components even in the presence of the catalyst. Obviously, the TST temperature of oil from the Usinskoye field should not be less than 250 °C, since at lower temperature, aquathermolysis is ineffective.

Results of the component composition in general indicate a positive effect of the influence of iron tallate in the thermal steam treatment of oil of the Usinskoye field at 300 °C. More than 3 times reduced the content of asphaltenes in the sample with the catalyst (4%) compared with the original oil, increased by 13% of the saturated hydrocarbon fraction and decreased by 10% the amount of resins in oil with iron thallate compared with the control experience.

### 2.4. Gas Chromatography—Mass Spectrometry of Saturated and Aromatic Hydrocarbons

Figure 10 shows GC-MS spectra of saturated hydrocarbons of the original bitumoid and bitumoids after TST at different temperatures and the presence of iron tallate by total ionic current (TIC).

As a result of thermal steam treatment at temperatures of 200 and 250 °C as well as in the presence of the catalyst iron tallate, the content of n- and iso-alkanes increases in comparison with the original oil and with the control experiments. The content of alkanes in the samples after TST at 300 °C increases by 8 times in comparison with the original oil, and the content of cycloalkanes in the sample with catalyst increases by 2 times in comparison with the control experiments. This could indicate the breaking not only of carbon heteroatom bonds (C-S, N, O) but also of C-C bonds (Figure 11).

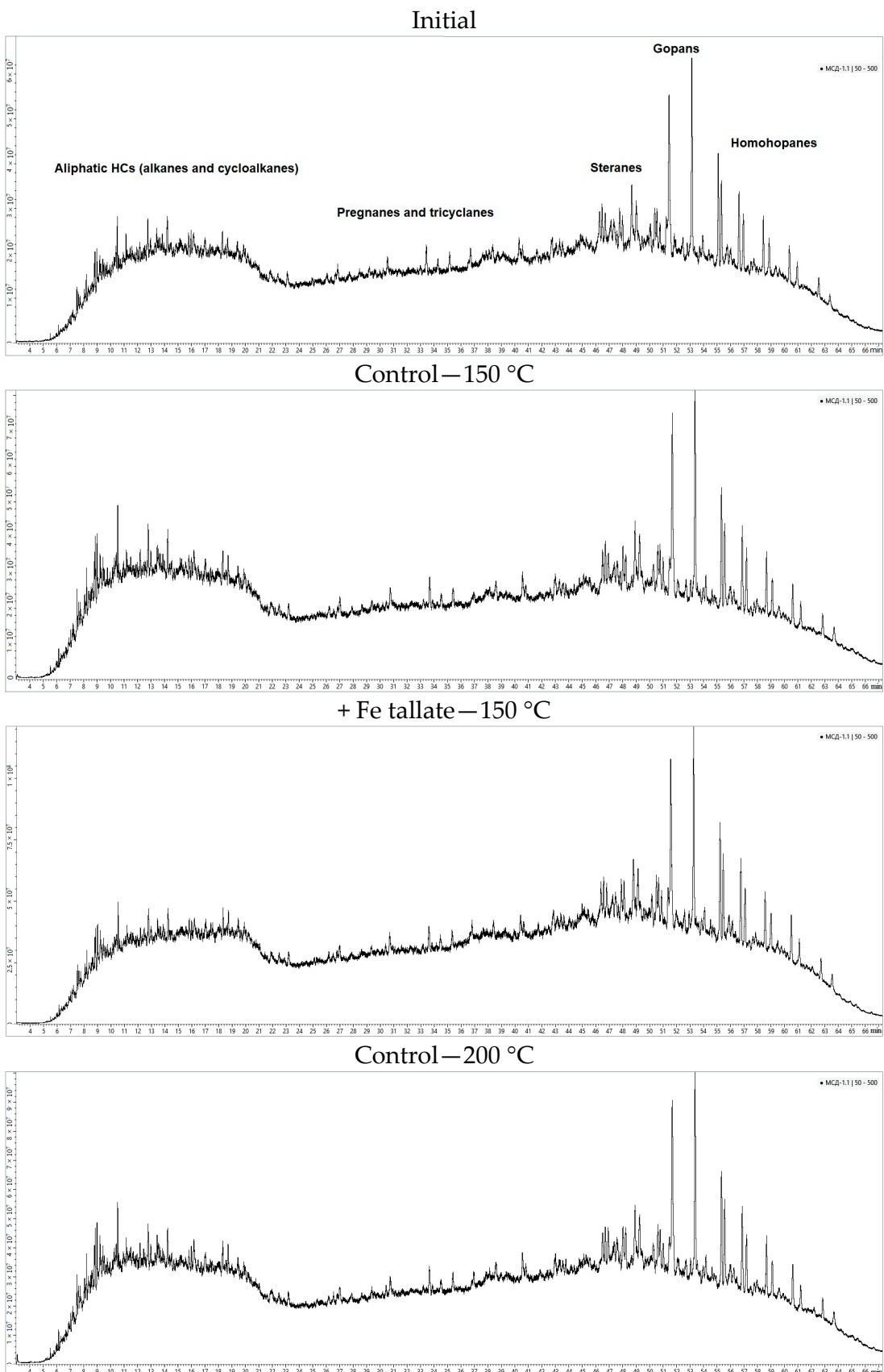

**Figure 10.** *Cont.*

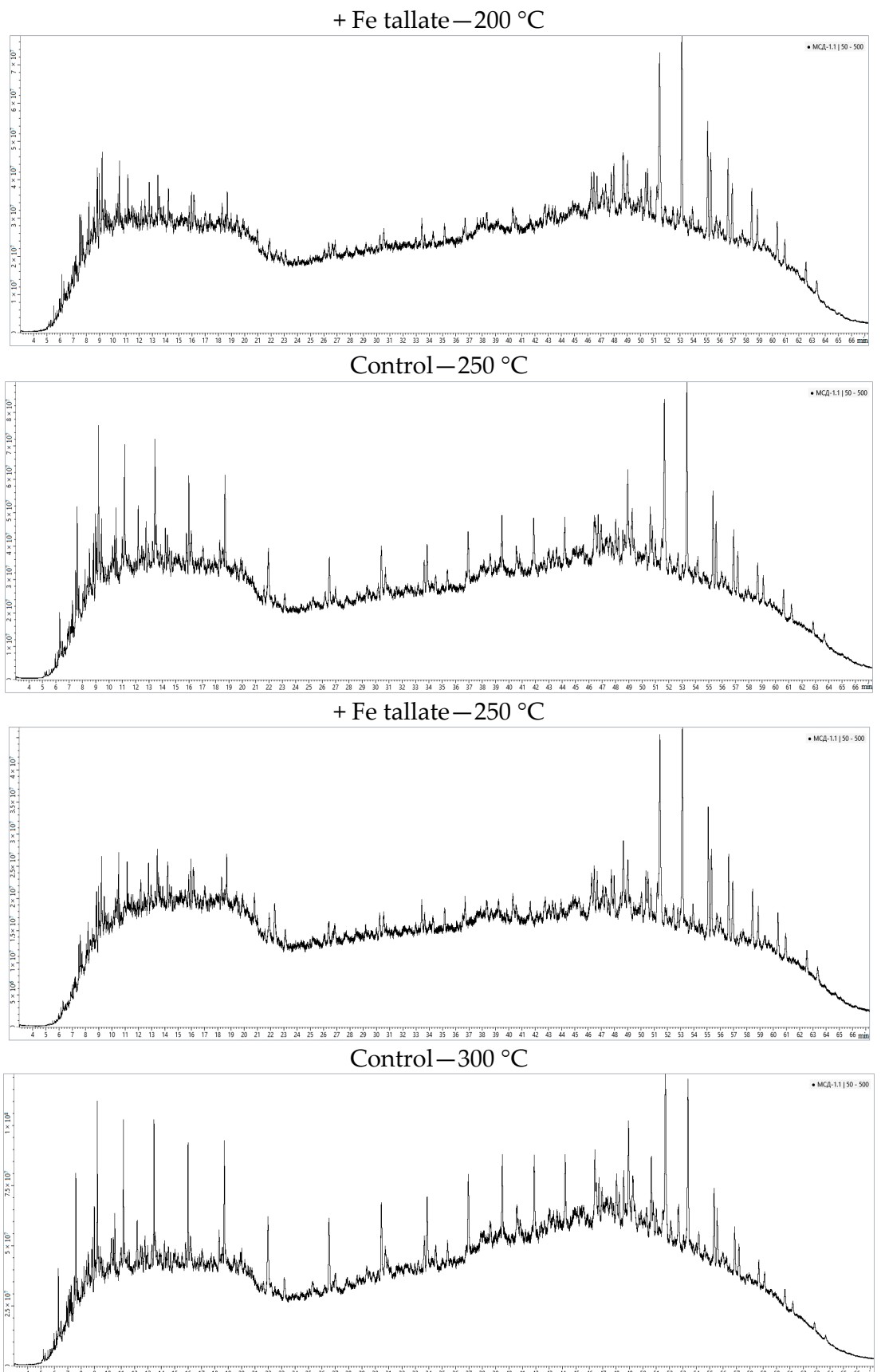

**Figure 10.** *Cont.*

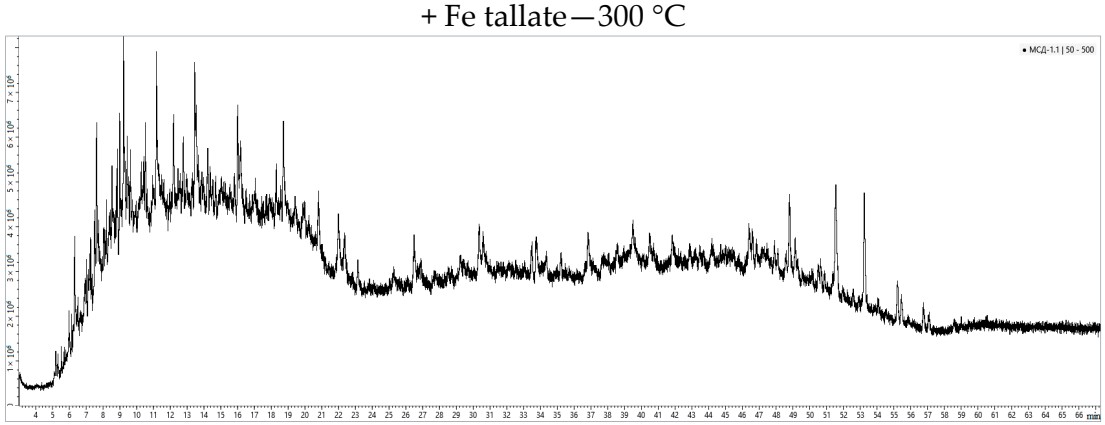

**Figure 10.** Total ionic current (TIC) chromatograms of the saturated fraction.

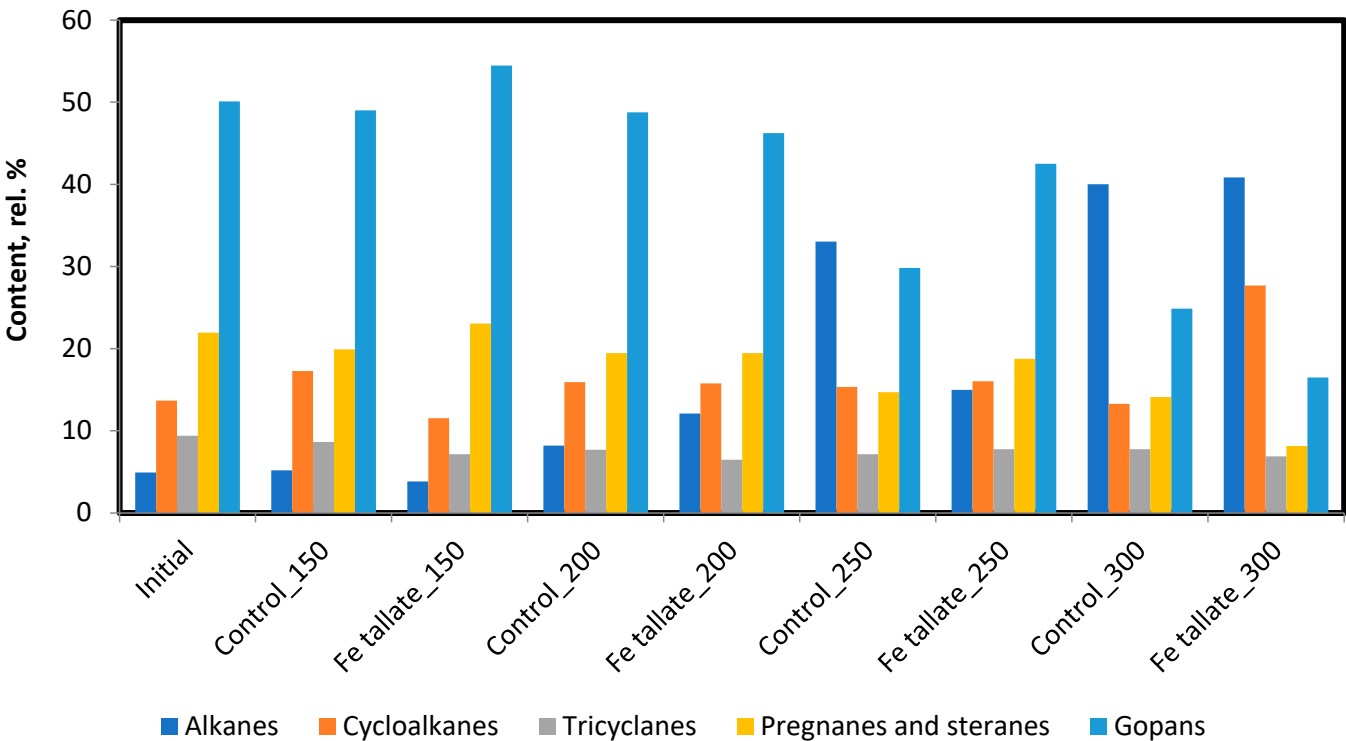

**Figure 11.** Relative content of identified compounds in saturated fraction.

Mass-fragmentograms of alkanes also confirm that at elevated temperatures (250 and 300 °C) and the use of catalysts increases the peaks corresponding to alkanes of normal and isoprenoid structure (Figure 12).

Figure 13 shows GC-MS spectra of aromatic hydrocarbons of the original bitumoid and bitumoids after TST at different temperatures and the presence of iron thallate by total ionic current (TIC).

The greatest changes on chromatograms of aromatic fraction were observed after catalyst influence together with high temperatures (250 and 300 °C): the relative content of naphthalene and its homologues increases, alkylbenzenes with shorter alkyl chain peaks appear, peaks corresponding to 1-methyl-7-isopropylphenantren and trimethylphenantren increases significantly.

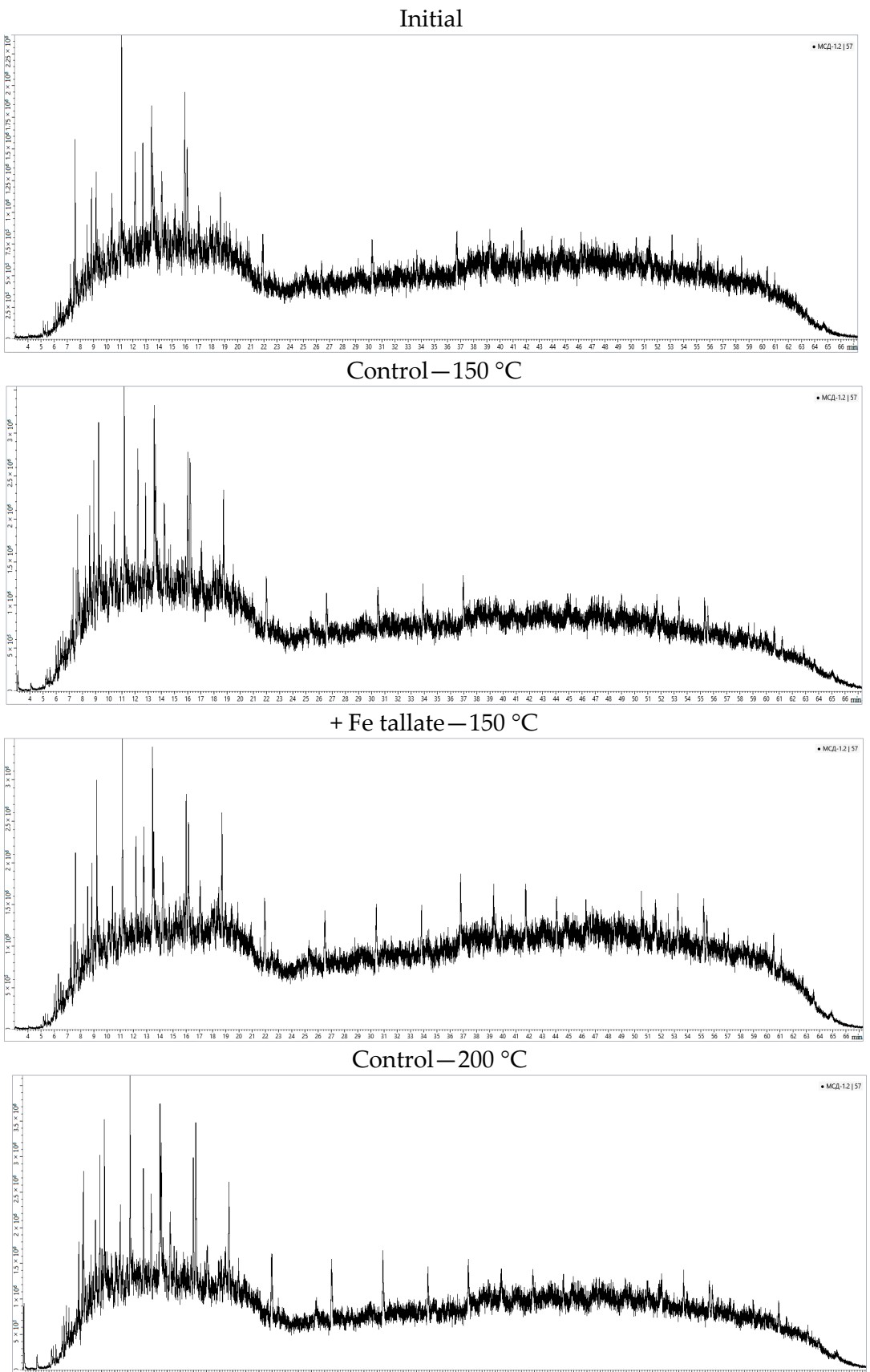

**Figure 12.** *Cont.*

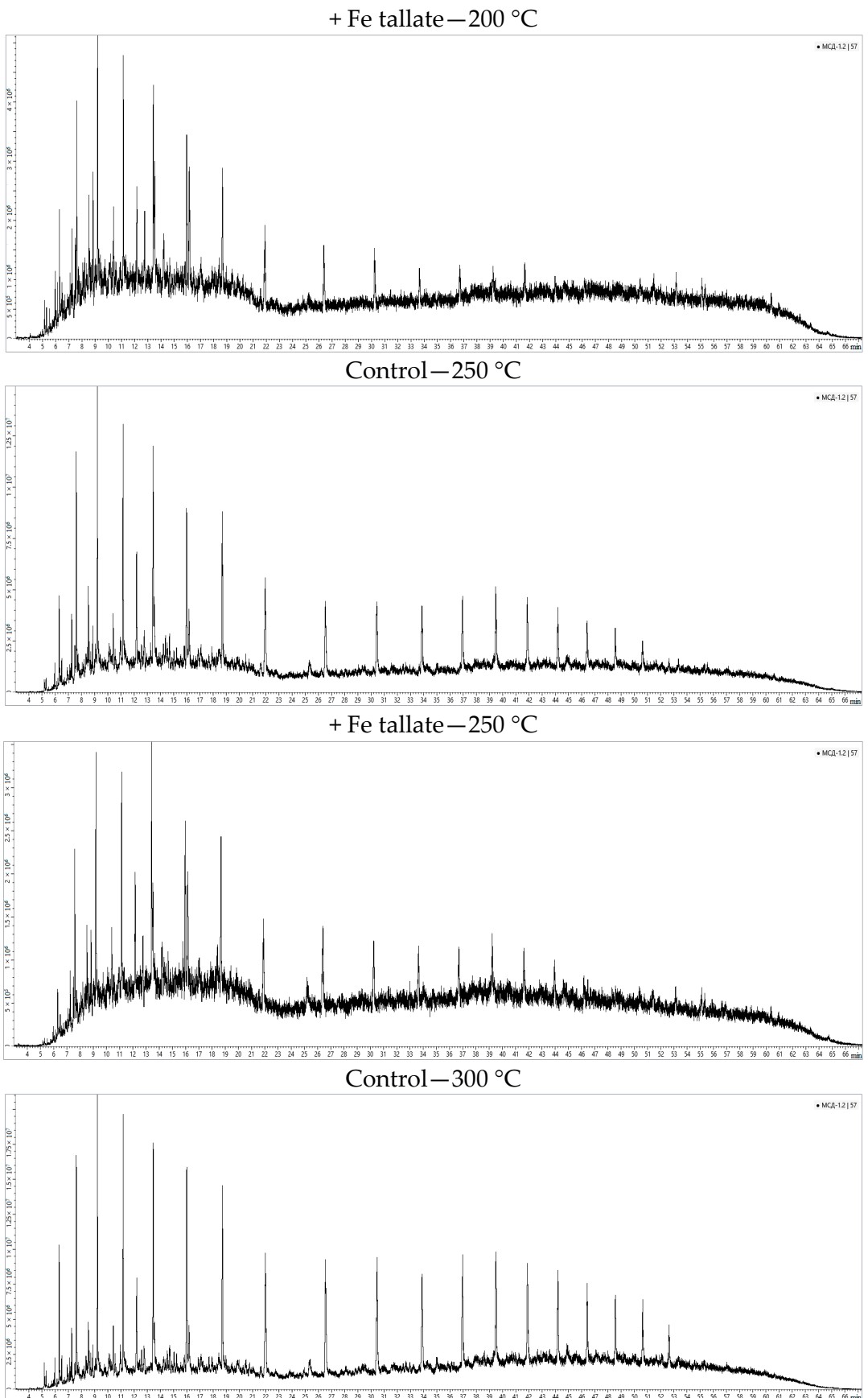

**Figure 12.** *Cont.*

### + Fe tallate—300 °C

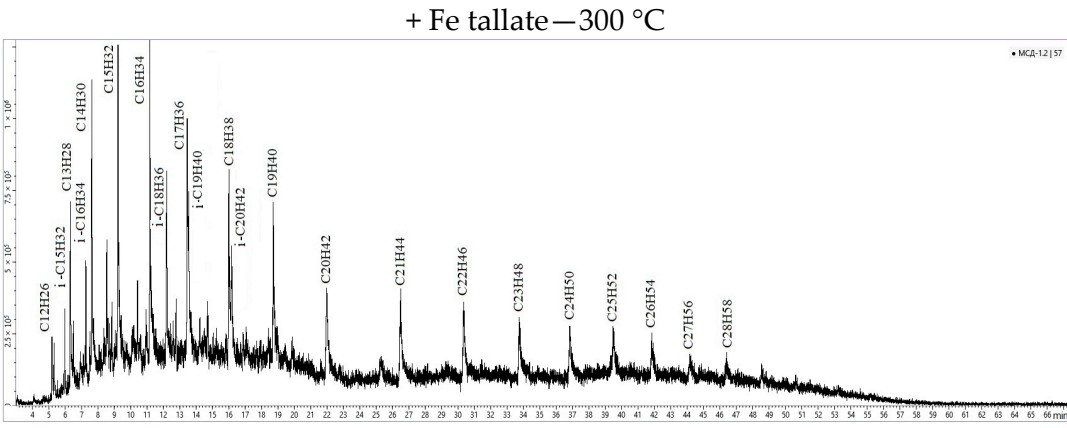

**Figure 12.** Mass-fragmentograms of alkanes (*m/z* = 57) from the saturated fraction.

### Initial

### Control—150 °C

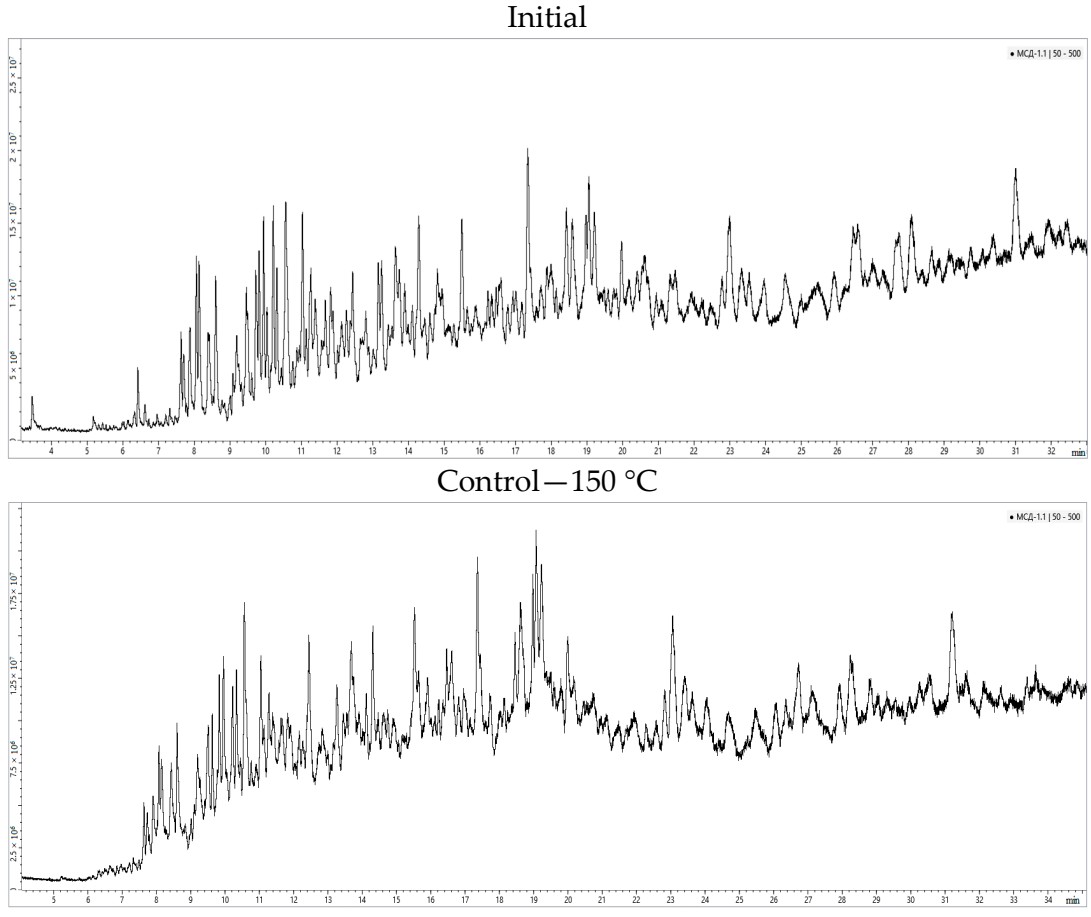

**Figure 13.** *Cont.*

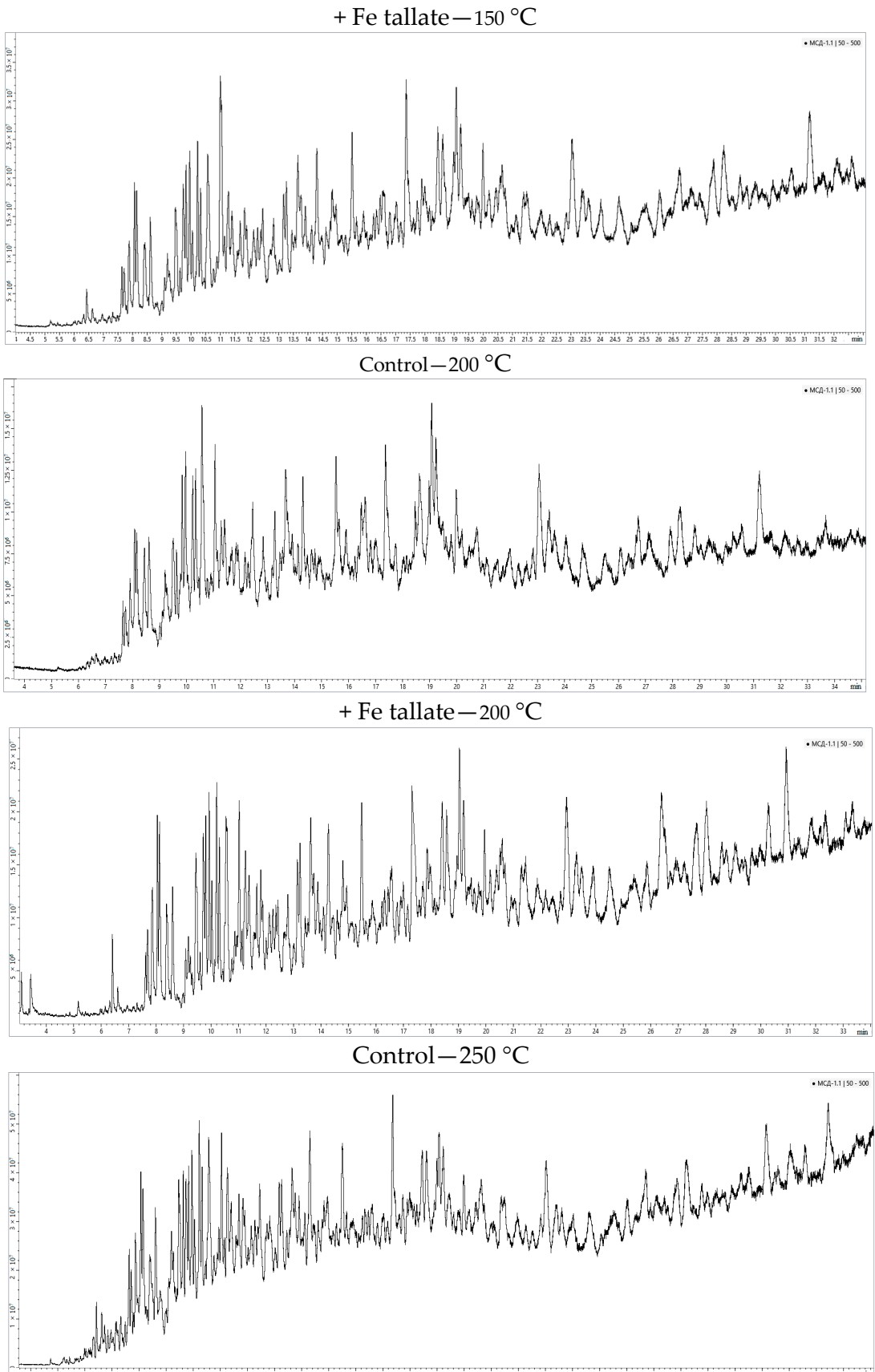

**Figure 13.** *Cont.*

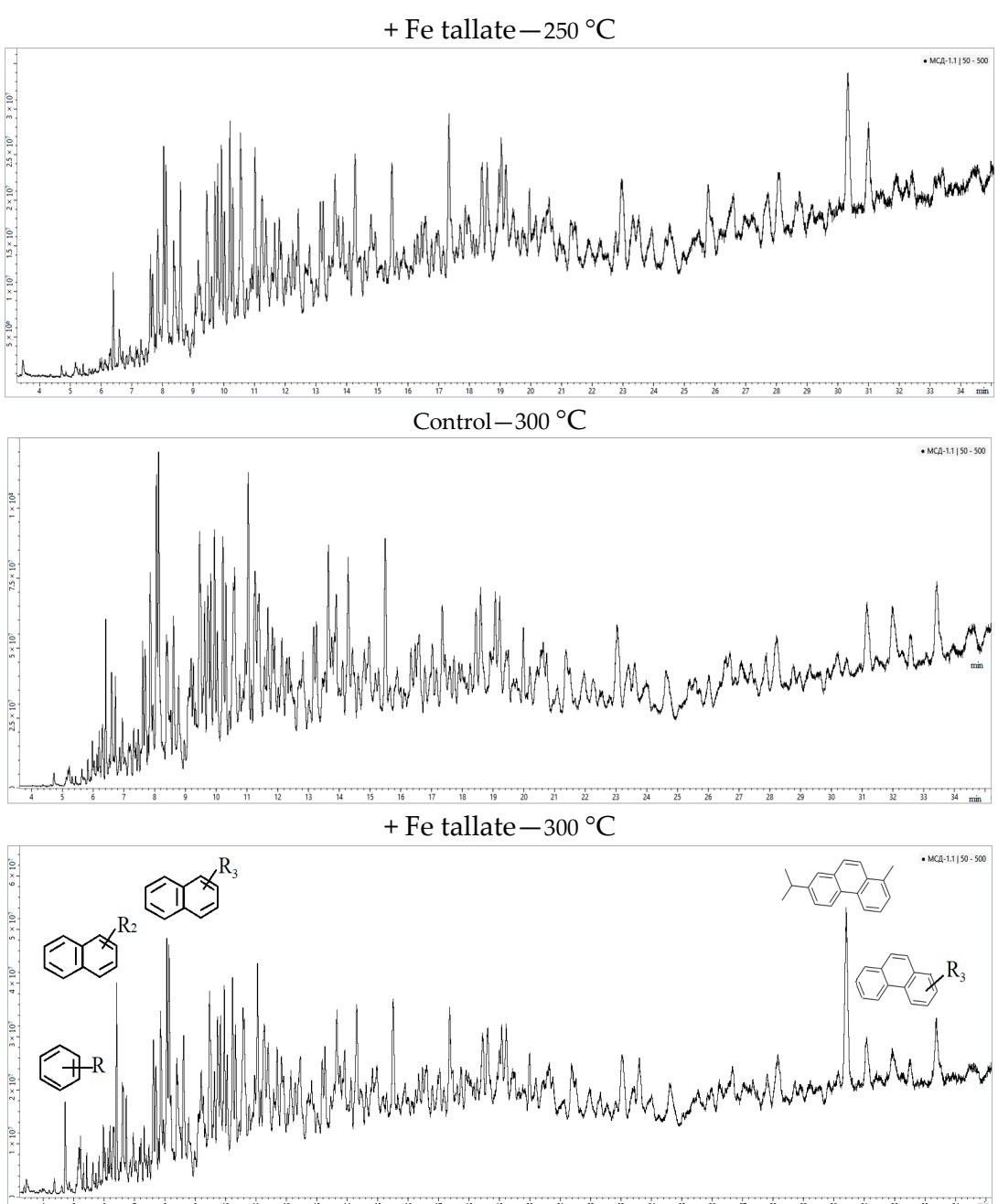

**Figure 13.** Total ionic current (TIC) chromatograms of aromatic fraction.

Figure 14 shows a diagram of the relative content of the identified aromatic HC compounds.

*2.5. Determination the Molecular Mass of Bitumoids*

The results of molecular mass measurements of oil samples are presented in Table 1.

From the obtained results, it can be concluded that the molecular mass of bitumoids decreases as a result of thermocatalytic treatment. During the transition from the control experiment to the catalytic treatment with hydrogen donor, the bituminous system reaches the most active state. By the term «activity» of an oil dispersed system is meant the change of its physicochemical properties under the influence of a unit of external influence ($r_{min}$, $h_{max}$) [38]. In our case, with the increase of the proportion of the catalyst precursor in the bitumoid, there was a phase inversion (r—particle radius of the asphaltene—particle radius of the asphaltene—disperse phasedisperse phase, h—thickness of the dispersion

medium—particle radius of the asphaltene—particle radius of the asphaltene—disperse phasedisperse phasemaltenes), i.e., a decrease in the particle size in oil dispersed system, which is the result of a redistribution of the high molecular mass ions towards the light fractions (saturated and aromatic HC).

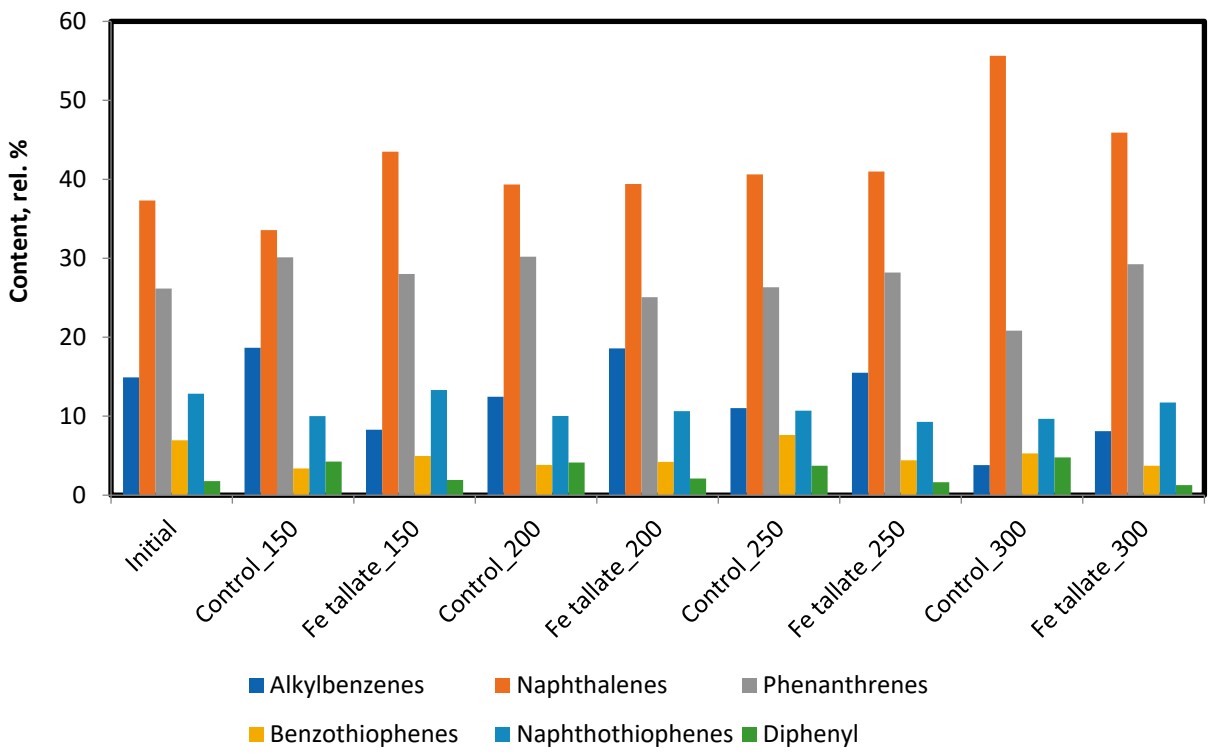

**Figure 14.** Relative content of identified compounds in the aromatic fraction.

**Table 1.** Results of molecular mass measurements of oil samples.

| Sample | Crystallization Temperature °C * | Mass of Sample, ($G_1$), g | Mass of Benzene ($G_2$), g | Depression of the Crystallization Temperature, (ΔT), °C | Molecular Mass of the Sample (M), g/mol |
|---|---|---|---|---|---|
| Benzene | 5.448 | | | | |
| Initial oil | 4.979 | 0.6374 | 12.4032 | 0.465 | 566 |
| Control 250 °C | 4.9 | 0.5266 | 11.9856 | 0.548 | 410.5 |
| Fe 250 °C (2%) | 4.987 | 0.5905 | 12.1076 | 0.458 | 545 |
| Fe 250 °C (4%) | 5.002 | 0.6090 | 12.0984 | 0.446 | 578 |
| Control 300 °C | 4.753 | 0.5318 | 11.9527 | 0.695 | 328 |
| Fe 300 °C (2%) | 4.964 | 0.4205 | 12.2296 | 0.481 | 366 |
| Fe 300 °C (4%) | 4.937 | 0.3312 | 11.6353 | 0.508 | 287 |

* True crystallization temperature is the highest temperature reached at which the liquid and solid phases are in equilibrium.

### 2.6. Studying the Rock Surface by Scanning Electron Microscopy

In the function of the catalyst, the active form was converted from the oil-soluble complex into complex iron sulfides, the content of which increases due to the reduction of the undecomposed part of the precursor. Images of catalyst particles on the rock taken by scanning electron microscope are shown in Figure 15.

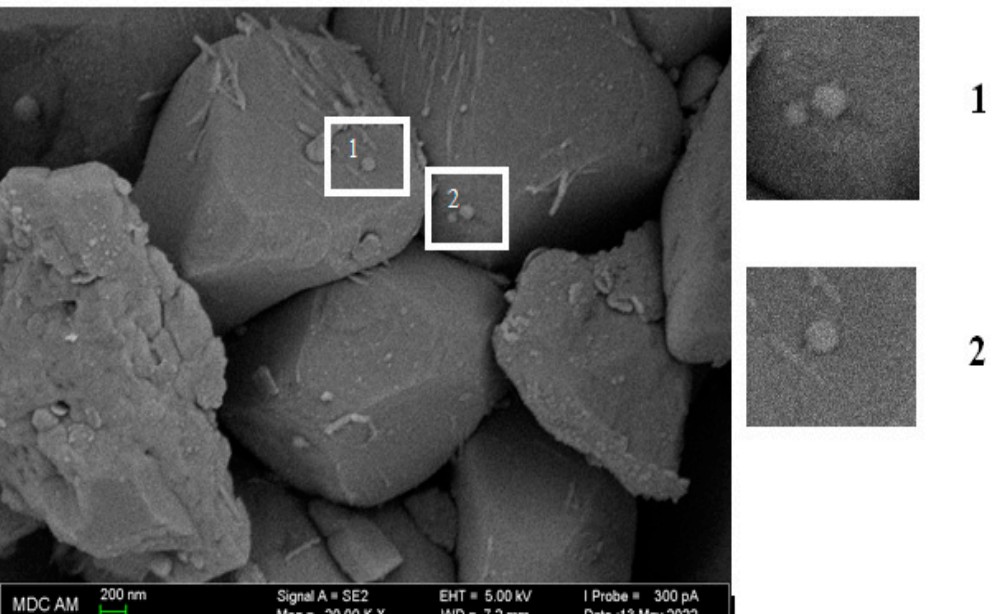

**Figure 15.** SEM images of catalyst particles on the rock.

According to SEM, the catalyst represents nanodispersed particles of size ≈60–80 nm.

Thus, the main process that provides oil transformation is similar to the hydrotreating process, in which sulfide catalysts on the carrier are highly effective. In the case of in situ aquathermolysis, the carrier is mineral grains of reservoir rock.

Transition metal atoms, in this case iron, bonded to the surface of metal sulfide crystallites via sulfide bridges, form the active catalytic centers of sulfide catalysts. These form edge centers that play a key role in the hydrocarbon conversion reactions that take place. In addition, it is believed that there is an exchange of sulfur in the catalyst with sulfur in the resins and asphaltenes—particle radius of the asphaltene—disperse phase the theory of dynamically created active centers, in which homolytic breaking of the C-S bond occurs after adsorption of hydrocarbons onto catalyst particles, and the free valencies become saturated with hydrogen. Thus, hydrocarbons with lower molecular mass were formed.

*2.7. Thermogravimetric Analysis for Determination of Residual Oil and Coke*

Figure 16 shows the results of determining the amount of coke deposited on the rock after extraction of the original bitumoid and the bitumoid after TST.

The TG-DTG curves indicate a significant role of the catalyst in oil conversion under reservoir conditions. Thus, a small amount of bitumoid, amounting to 0.22%, remains in the original rock sample after extraction. By increasing the temperature of thermal steam treatment of the rock, the subsequent extraction leads to an increase in the bitumen content from 0.18% (150 °C)–0.25% (200 °C)–1.1% (250 °C) to 1.76% (300 °C). In the presence of a catalyst and a hydrogen donor, this dependence was slightly lower and changed in the following series: from 0.17% (150 °C)–0.15% (200 °C)–0.16% (250 °C) to 1.21% (300 °C). The largest amount of coke was observed in the control sample at 300 °C, which indicates insufficient enrichment of the oil, condensation and densification of the aromatic rings of the resinous asphaltenes, in contrast to the sample with iron thallate, where the catalyst provides detachment of the alkyl substituents from the highly condensed components of the oil and regrouping of the component composition.

Based on the results of thermal analysis of the extracted samples of the source rock and after thermocatalytic treatment, the material balance of organic matter (OM) was calculated (Table 2). The content of coke was calculated assuming that the coke substances consist entirely of carbon and that the deviation from the average content of organic matter does not exceed 2%.

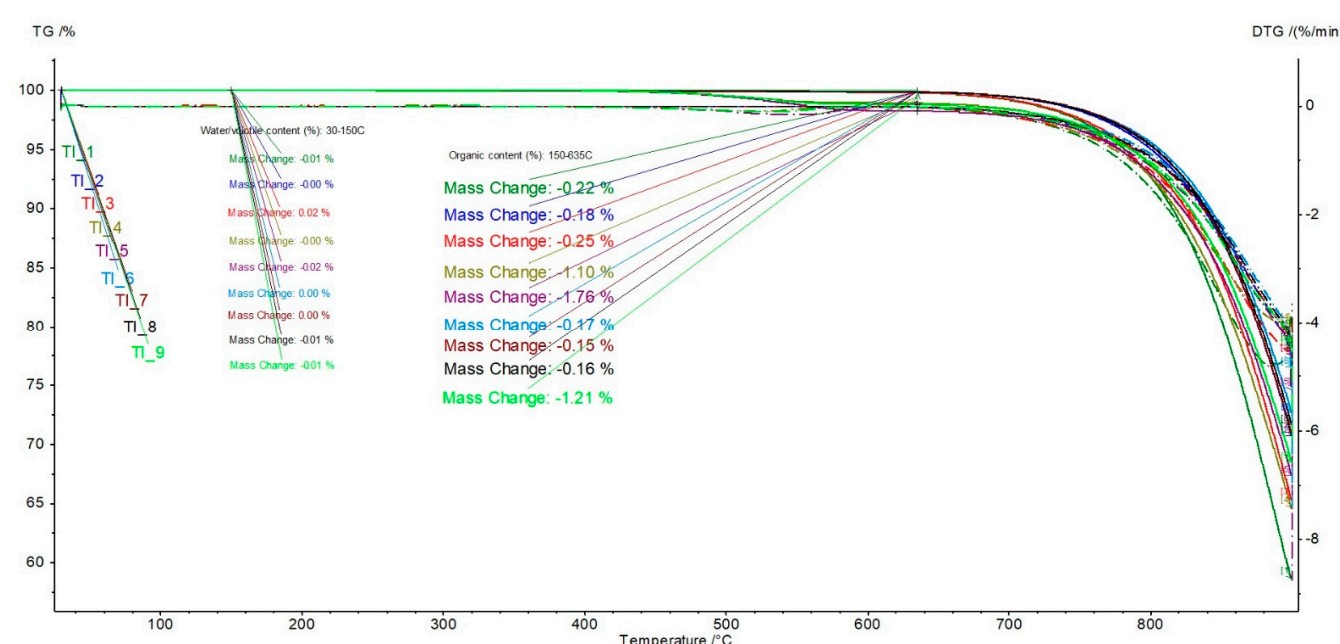

**Figure 16.** Thermogravimetric analysis of the Usinskoye deposit rocks after extraction.

**Table 2.** Material balance for organic matter.

| Samples | | Catalyst | Gases | Oil Content, Mass % | Mass Loss at Oxidation Temperature of OM | Total |
|---|---|---|---|---|---|---|
| Initial rock | | - | - | 7.13 | 0.22 | 7.35 |
| Conditions | 150-24 h | - | 0.049 | 8.96 | 0.18 | 9.189 |
| | 200-24 h | - | 0.023 | 8.28 | 0.25 | 8.553 |
| | 250-24 h | - | 0.231 | 7.22 | 1.1 | 8.551 |
| | 300-24 h | - | 0.703 | 3.46 | 1.76 | 5.923 |
| | 150-24 h | Fe | 0.033 | 7.96 | 0.17 | 8.163 |
| | 200-24 h | Fe | 0.109 | 6.97 | 0.15 | 7.229 |
| | 250-24 h | Fe | 0.146 | 6.82 | 0.16 | 7.126 |
| | 300-24 h | Fe | 0.446 | 4.96 | 1.21 | 6.616 |

## 3. Materials and Methods

*General information about the oil field.* The Usinskoye oilfield is located in the Usinsk region of the Komi Republic and is confined to one of the local structures of the Kolvin­sky megalval, the largest oil and gas accumulation zone, where such large oilfields as Voseyskoye, Kharyaginskoye, Yuzhno-Khylchuyuskoye are located. Moreover, these oil fields are located in a wide stratigraphic spectrum of petroleum deposits, ranging from Lower Devonian to Triassic. Geographically, the area of the field is part of the Pechora Plain within the Q-40 XVI and belongs to the watershed of the lower reaches of the Kolva River, the right tributary of the Usa River, which flows into the Pechora.

The geological cross-section of the Usinskoye field was studied from the Silurian to the Quaternary deposits, with the deepest borehole of the Usinskoye survey (borehole No. 37) revealing Lower Silurian deposits at a depth of 5005 m.

Tectonically, the Usinskoye field is confined to the anticline of the same name, which complicates the southern end of the Colvin megavalve in the Timan Pechora Province.

The Permo-Carboniferous deposit of the Usinskoye field is one of the largest and most complex targets currently being developed in the Timan-Pechora Province. The deposit lies at a depth of 1100–1500 m and contains unusually viscous oil (710 mPa·s)

in cavernous-porous fractured carbonates of Lower Permian, Upper Carboniferous and Middle Carboniferous age. The deposit is covered by a layer of Upper Permian siltstones, mudstones and clays.

The deposit was vaulted, massive, dimensions 16.0 km × 8.5 km, oil bearing capacity 356 m, thickness of oil saturated limestones varied from 0 on the contour to 172 m in the central part of the object.

The high total porosity of the reservoirs (0.182) is mainly associated with secondary pores of leaching and diagenetic dolomitization-percrystallization, ranging in size from 0.1–1.0 mm, variably configured and irregularly located in the rock, often near fractures and styrofoam joints, communicating with each other and with cavernous channels 100 to 300 μm wide, up to 10 mm long, and microcracks. Overall, according to the logging data for oil-bearing reservoirs, the porosity of the formation was 21.3%, after the core sample 20.5%, after the thermal impact on the formation 20% and 18.2%, respectively. According to the results of the study of the azimuthal anisotropy of the filtration properties and its relationship with the fracturing of the Permo-Carboniferous strata (more than 100 measurements were made in borehole 46), a close relationship was found between the values of the gas permeability of the rock and the azimuthal orientation of the fractures. Most fractures (80%) are confined to the 45–95° strike azimuth. The gas permeability values measured along the fracture azimuth (up to 1 μm$^2$) are several times higher than the gas permeability in the perpendicular direction (from $10^{-3}$ to $10^{-1}$ μm$^2$).

The average arithmetic residual oil saturation of the reservoir rock in the core of the whole reservoir is 61% (in some wells it reaches 80%) [39].

The subject of the study—the oil-bearing rock of the 5317 well (sampling interval—1303.0–1306.6 m).

### 3.1. Analysis of the Structural Composition of the Source Rock

The phase composition of the source rock sample from the Usinskoye field was studied by X-ray powder diffractometry using a MiniFlex 600 diffractometer (Rigaku, Tokyo, Japan) equipped with a high-speed D/teX detector. Measurements were performed using CuKa radiation (40 kV, 15 mA), the results were obtained at room temperature in the range of angles 2θ from 3 to 100 in 0.02 increments and exposure times at each point 0.24 s without rotation.

### 3.2. Laboratory Modeling of Hydrothermal Catalytic Processes

Simulation of thermal steam treatment of a core sample of the Usinskoye field was performed in a Parr Instruments autoclave reactor at temperatures of 150, 200, 250, and 300 °C for 24 h in a nitrogen atmosphere. In addition, the catalyst was injected into hydrogen donors in the form of a 50% solution. Nefras C4-155/205, a mixture of naphtheno-aromatic compounds, was chosen as the hydrogen donor. The amount chosen was 2 and 4 wt.% per oil of each reagent. The crushed core, catalyst solution and distilled water were immersed in the beaker of the autoclave. The rock:water ratio was 10:1. After catalytic aquathermolysis, the effect of the catalyst on the superviscous oil was studied by the methods described below.

### 3.3. Gas Phase Chromatographic Analysis

Gas composition was investigated after thermocatalytic treatment of core with the help of chromatograph (Chromatec-Crystall 5000, Yoshkar-Ola, Russia). For the analysis, a sample of the gas phase was taken using a special outlet in the autoclave cover into the hose leading to the gas chromatograph. The chromatograph column was purged with aquathermolysis gases for saturation. The gases were separated using a 100 m long, 0.25 mm diameter capillary column. At the following temperatures the chromatograms were taken: 90 deg for 4 min and then heated 10 deg/min to 250 °C. The evaporator temperature was 250 °C. The carrier gas was helium and the flow rate was 15 mL/min. Compounds were identified using the NIST digital library and literature data [40].

### 3.4. Preparation and Performance of Hot Solvent Extraction with Determination of Bitumoid Content

In order to obtain reliable results, subsequent laboratory-analytical methods, it is necessary to carry out qualitative sample preparation of the studied rock samples. The first stage of sample preparation of rock samples is extraction—treatment of rocks with organic solvents in order to extract soluble substances. The extraction process mechanism includes not only dissolution, but also desorption, which is slower than the dissolution itself, which somewhat limits the solvent. In petrochemistry, solvents with low boiling points (chloroform, benzene, etc.) are used; the extracted organic substances are now called bitumoids.

The extraction procedure consists of the following: the rock recoverable was placed in a sleeve made of filter paper. Then, the empty and filled sleeves were weighed on analytical scales with an accuracy of 0.01 g. The sample size was selected on the basis of the preliminary analysis of organic carbon ($C_{org}$). If the content of bitumen in the rock is about 0.01% mass of the sample should be 500 g rock, if 0.02–0.1%—200–300 g, etc. Bitumen extracted using «triplet» which is a mixture of equal amounts of benzene, isopropyl alcohol and chloroform, providing the most complete extraction of bitumen components of different chemical nature. The casing with the rock is placed and filled with solvent so that 1.5 of the extractor drain enters the flask through the siphon tube. This ensures that the flask contains at least 1/4 of the solvent volume, while the extractor is filled to the level of the siphon tube. At the end of the extraction, all the solvent was distilled at a Hei-VAP Precision rotary evaporator at 40 °C and a pressure of 0.013 MPa.

To determine the content, it was necessary after bringing the mass of the flask with bitumen to a constant value to determine its mass and yield per rock, which was calculated by the formula:

$$\textit{Extract} \text{ (mass \%)} = a \times 100/A$$

where *Extract* is the content of bitumoid in the rock, %; *a*—is the weight of bitumoid, g; *A*—is the mass of the rock taken for extraction, g.

### 3.5. Determination of Viscosity and Rheological Curves

Measurements of the bitumoid rheological parameters from the Usinskoye field were carried out using a plate/plate system of Anton Paar's MCR 302 automatic rheometer (Graz, Austria). The measurements were performed at a temperature of 25 °C, and the sample volume was 0.49 mL. Rheological parameters were determined in rotational mode, with the shear rate set from 1 to 100 s$^{-1}$. It took an average of five minutes to measure each point at one shear rate. In addition, at low shear rates, each point was measured for 15 min.

### 3.6. Determination of the Group Composition of Oil

After extraction of the oil-saturated core and subsequent solvent stripping, the group composition was determined. The oil was separated by sedimentation in solutions of malts and asphaltenes for 24 h and filtered. The malts were separated into 3 groups by the method of SARA -analysis (saturated and aromatic hydrocarbons, resins) according to GOST 32269-2013.

In this context, fractionation was performed on a chromatographic glass column followed by elution with aliphatic (n-hexane), aromatic hydrocarbons (toluene) and a mixture of aromatic and polar solvents (toluene + isopropanol 3:1) from an adsorbent previously dehydrated at 450° C for 3 h [41,42].

### 3.7. Gas Chromatography—Mass Spectrometry of Saturated and Aromatic HCs

GC/MS analysis of saturated hydrocarbons fractions from the original oil samples and after water-thermolysis with different additives was carried out on chromatograph Chromatec-Crystall 5000.2 (Chromatec, Yoshkar-Ola, Russia) with the mass-spectrometric detector 214.2.840.083-10 (ion source ADVIS) with the computer processing of data on ions *m/z* 57 for alkanes. For aromatic HC, we used computer processing of data on ions *m/z*

133 (alkylbenzenes), *m/z* 128, 142, 156 (naphthalenes), *m/z* 178, 192, 206 (phenanthrenes), *m/z* 132, 146, 160 (tetralenes).

### 3.8. Determination of the Molecular Mass of Bitumoids by Cryoscopy in Benzene

The method is based on measuring the temperature depression—the temperature difference of crystallization of pure solvent (benzene) and oil sample solution, which is then used to calculate the molecular mass of the sample. Measurements were carried out on the KRION-1 unit (Termex, Tomsk, Russia). A low-temperature liquid thermostat KRIO-VT-12 was used to create and maintain test temperatures.

### 3.9. Studying the Rock Surface by Scanning Electron Microscopy

*Preparation technique*. In order to obtain high-resolution microphotographs, the objects were previously sprayed with 80/20 Au/Pd alloy, and the deposits were excluded during quantitative and qualitative analysis. In addition, the specimens mounted on a support were placed in an electron microscope chamber. Probing was performed on selected areas.

*Measurement technique.* The examinations were performed using a Merlin (Carl Zeiss) scanning electron microscope with autoemission. The microscope was equipped with an AZtec X-Max energy dispersive spectrometer (Oxford Instruments NanoAnalysis & Asylum Research, High Wycombe, UK). The resolution of the spectrometer was 127 eV, and surface morphology was recorded at an accelerating voltage of 5 keV to improve the depth of field of the images. Elemental analysis was also performed at an accelerating voltage of 20 keV and a working distance of 9 mm, which avoids minimal errors because the probing depth for elemental analysis is about 1 $\mu$m.

### 3.10. Thermogravimetric Analysis (TGA) for Determination of Residual Oil and Coke

Thermogravimetric experiments under atmospheric conditions were performed on a precision TG209 F1 Libra thermogravimeter (Netzsch GmbH, Selb, Germany) in combination with an Alpha FTIR spectrometer (Bruker GmbH, karlsruhe, Germany) in mass signal registration mode according to ASTM E 2105-00 (or GOST 57988-2017). These experiments were performed in a dynamic air/nitrogen environment at a linear heating rate of 10 K/min to 900 0C in 85 $\mu$L corundum crucibles. The instrument was preheated for one week to obtain a stable signal from the thermo-microbalance, and the temperature of the circulator connected to the instrument was 25 °C. The gases used for the experiments have a high degree of purity, which is an important criterion for a successful experiment. For additional purification, a replaceable system of low-pressure filters (up to 10 bar) was used.

## 4. Conclusions

In this study, we synthesized an iron-based oil soluble catalyst as a precursor and investigated its effect on aquathermolysis of heavy oil from the Usinskoye field at different temperatures (150, 200, 250, 300 °C) for 24 h.

As a result of control experiments and with the use of iron thallate obtained data indicating that:

- at 300 °C with the catalyst (2%), the content of saturated hydrocarbons (HC) increases by 10% (compared to the control experiment) and the content of aromatic HC increases by 5% (compared to the original oil);
- asphaltenes content decreases by 48% (comparison with 300 °C Fe 2% experience) and by 71% (comparison with the original oil) at 300 °C Fe with a catalyst (4%);
- viscosity is reduced by 50% at the temperature 300 °C in comparison with the experience of the control;
- in the samples after TST at 300 °C, the content of alkanes increases 8-fold compared to the initial oil, and the content of cycloalkanes in the sample with the catalyst increases 2-fold compared to the control experience. This could indicate that not only carbon-heteroatom bonds (C-S,N,O) but also the breakage of C-C bonds;

- the increase in the concentration of iron thallate at TST 300 °C led to a decrease in the molecular mass of the oil compared to the control experiment;
- according to SEM, the catalyst is nanodisperse particles of size ≈60–80 nm and adsorption on the rock surface takes place, the catalyst removal will take place to a minimal extent;
- the highest amount of coke was observed in the control sample at 300 °C (1.76%), indicating insufficient enrichment of oil, condensation and compaction of aromatic rings of resinous-asphaltene substances in contrast to the sample with iron thallate (1.21%).

Thus, in summary, energy efficiency is increased by steam-thermal methods for the extraction of this type of unconventional resources, with the research conducted demonstrating the possibility of wide application of catalytic compositions in porous mineral media of carbonate reservoir rocks.

**Author Contributions:** Conceptualization, I.I.M.; data curation, R.E.M.; investigation, I.I.M., A.V.L., R.E.M., A.A.A. and D.A.E.; methodology, I.I.M., A.V.L., A.A.A. and D.A.E.; project administration, O.V.S.; supervision, O.V.S.; validation, B.A.; visualization, A.V.V.; writing—original draft, I.I.M.; writing—review & editing, A.V.V. All authors have read and agreed to the published version of the manuscript.

**Funding:** This work was supported by the Ministry of Science and Higher Education of the Russian Federation under agreement No. 075-15-2022-299 within the framework of the development program for a world-class Research Center «Efficient development of the global liquid hydrocarbon reserves».

**Data Availability Statement:** Not applicable.

**Acknowledgments:** We would like to express our gratitude to the Interdisciplinary Center for Analytical Microscopy, Kazan Federal University, for their help in providing research on rock microscopy.

**Conflicts of Interest:** The authors declare no conflict of interest.

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
