# Peer review of "Study of the Hydrothermal-Catalytic Influence on the Oil-Bearing Rocks of the Usinskoye Oil Field"

_catalysts, doi:10.3390/catal12101268_

Round 1

Reviewer 1 Report

This manuscript investigated the hydrothermal-catalytic cracking of heavy oil by using iron-based petroleum soluble catalyst. The effect of temperature on the upgraded heavy oil properties and composition was analyzed. Some results are obtained and the discussed is well. This manuscript can be accepted after considering the following issues:

1.      The introduction is too long. It should be more concise and show the main research purpose of this work. Moreover, the novelty of this paper should be added.

2.      This manuscript seems a experimental report. More discussion should provided.

3.      The XRD patterns of fresh catalyst particles on the rock should be added in Figure 1.

4.      As shown in Table 1, the crystallization temperature is shown. How to determine the crystallization temperature?

5.      Figure 15 shows the fresh catalyst particles on the rock. To better understand the effect of catalyst, catalyst after reaction should also be provided.

Author Response

1. The introduction is too long. It should be more concise and show the main research purpose of this work. Moreover, the novelty of this paper should be added.

Actually, we don't find the introduction too long. It consists of concrete work on the enrichment of heavy oils with iron-containing catalysts. Moreover, the scientific novelty of this work consists in the fact that the conducted researches prove the possibility of wide application of catalytic compositions in porous mineral environments of carbonate reservoir rocks. Moreover, the Usinskoye field is one of the examples of carbonate deposits.

2. This manuscript seems a experimental report. More discussion should provided.

The results are described in sufficient detail. It may be that the article is a bit overloaded mainly with the results of the chromatograms.

3. The XRD patterns of fresh catalyst particles on the rock should be added in Figure 1.

Figure 1 shows an X-ray diffractogram of the original rock sample after extraction to show that the rock was carbonate. X-ray diffractograms of catalyst particles on the rock were not taken due to impossibility to extract such amount of catalyst that was used in the experiments.

4. As shown in Table 1, the crystallization temperature is shown. How to determine the crystallization temperature?

The unit operating principle is based on measuring the temperature depression which is the difference between the crystallization temperature of a pure solvent and a solution of the substance under test, which is then used to calculate the molecular weight of the substance under test using the formula:

M=5.12*(1/ΔT)*(G1/G2)*1000

где M — is the molecular mass of the sample, g/mol;

5.12 — cryoscopic constant of benzene, °Ð¡-g/mol;

ΔT — temperature depression, °Ð¡;

G1 — oil sample weight oil, g;

G2 — benzene sample wieght, g

At first, the freezing temperature of pure benzene was determined. in order to do that, 20-25 cm3 of benzene was poured into a test tube and an agitator and a thermometer were inserted so that the ball of the latter was in the middle of the liquid. Next, the test tube with the sleeve was placed in a cryostat (the temperature of the mixture was maintained at 2-3 °C). By continuously stirring the benzene, observing on the thermometer the temperature decrease. Thus, the highest temperature shown by the thermometer after the beginning of crystallization was the freezing point of the liquid. Also, by counting the divisions of the thermometer with an accuracy of up to 0.01 °C. Subsequently, the freezing temperature of oil solutions in benzene was determined. Solutions of different concentrations, from 0.2 to 1.0 %, were prepared. To do this, into conical flasks with fused stoppers three samples (with an error of not more than 0.0002 g) of oil are taken: 0.16-0.25, 0.25-0.35 and 0.35-0.50 g. 20-25 cm3 of benzene is poured into each flask and weighed again, thus establishing the exact mass of benzene taken. For each solution the freezing point is determined in the same way as the freezing point of the pure solvent.

5. Figure 15 shows the fresh catalyst particles on the rock. To better understand the effect of catalyst, catalyst after reaction should also be provided.

Figure 15 shows catalyst particles on the rock after hydrothermal-catalytic enrichment at 300 °C.

Reviewer 2 Report

In this work synthesis of an oil-soluble iron-based catalyst precursor was carried out 9 and its efficiency was tested in laboratory simulation of the aqua-thermolysis process at different 10 temperatures. The rocks of the Usinskoye field of Permian deposits of the Komi Republic obtained by steam-gravity drainage, the iron-based catalyst precursor and the products of non-catalytic and catalytic aqua-thermolysis were used as the object of study. It was shown that the content of alkanes in the samples after thermal steam treatment (TST) at 300 °C increased by a factor of 8 compared to the original oil, and the content of cycloalkanes in the sample with the catalyst increased by a factor of 2 compared to the control experience. This may indicate that not only the carbon-heteroatom bonds (C-S, N, O), but also the C-C bonds were broken. It also shows that increasing the iron tallate concentration at TST 300 °C leads to a decrease in the molecular mass of the oil compared to the control experiment. According to SEM, the catalyst represents nano-dispersed particles with a size of ≈ 60-80 nm that are adsorbed on the rock surface, catalyst removal occurs to 20 a minimum extent. The article is quite interesting; it includes enough work and adds some new information especially on the hydrothermal-catalytic influence on the oil-bearing rocks of the Usinskoye oil field. Moreover, it adheres to the journal’s standards as Catalysts is an international open access journal of catalysts & catalyzed reactions. Thus, my recommendation for the manuscript is to be accepted after minor revision.

SPECIFIC COMMENTS:

- English should be improved by a native speaker

- The innovation of the presented work should be more clearly discussed

Author Response

The scientific novelty of this work consists in the fact that the conducted researches prove the possibility of wide application of catalytic compositions in porous mineral environments of carbonate reservoir rocks. Moreover, the Usinskoye field is one of the examples of carbonate deposits.
